# Learning Energy-Based Models by Diffusion Recovery Likelihood

**Ruiqi Gao**
UCLA
ruiqigao@ucla.edu

**Yang Song**
Stanford University
yangsong@cs.stanford.edu

**Ben Poole**
Google Brain
pooleb@google.com

**Ying Nian Wu**
UCLA
ywu@stat.ucla.edu

**Diederik P. Kingma**
Google Brain
durk@google.com

## Abstract

While energy-based models (EBMs) exhibit a number of desirable properties, training and sampling on high-dimensional datasets remains challenging. Inspired by recent progress on diffusion probabilistic models, we present a diffusion recovery likelihood method to tractably learn and sample from a sequence of EBMs trained on increasingly noisy versions of a dataset. Each EBM is trained with recovery likelihood, which maximizes the conditional probability of the data at a certain noise level given their noisy versions at a higher noise level. Optimizing recovery likelihood is more tractable than marginal likelihood, as sampling from the conditional distributions is much easier than sampling from the marginal distributions. After training, synthesized images can be generated by the sampling process that initializes from Gaussian white noise distribution and progressively samples the conditional distributions at decreasingly lower noise levels. Our method generates high fidelity samples on various image datasets. On unconditional CIFAR-10 our method achieves FID 9.58 and inception score 8.30, superior to the majority of GANs. Moreover, we demonstrate that unlike previous work on EBMs, our long-run MCMC samples from the conditional distributions do not diverge and still represent realistic images, allowing us to accurately estimate the normalized density of data even for high-dimensional datasets. Our implementation is available at https://github.com/ruiqigao/recovery_likelihood.

## 1 Introduction

EBMs (LeCun et al., 2006; Ngiam et al., 2011; Kim & Bengio, 2016; Zhao et al., 2016; Goyal et al., 2017; Xie et al., 2016b; Finn et al., 2016; Gao et al., 2018; Kumar et al., 2019; Nijkamp et al., 2019b; Du & Mordatch, 2019; Grathwohl et al., 2019; Desjardins et al., 2011; Gao et al., 2020; Che et al., 2020; Grathwohl et al., 2020; Qiu et al., 2019; Rhodes et al., 2020) are an appealing class of probabilistic models, which can be viewed as generative versions of discriminators (Jin et al., 2017; Lazarow et al., 2017; Lee et al., 2018; Grathwohl et al., 2020), yet can be learned from unlabeled data. Despite a number of desirable properties, two challenges remain for training EBMs on high-dimensional datasets. First, learning EBMs by maximum likelihood requires Markov Chain Monte Carlo (MCMC) to generate samples from the model, which can be extremely expensive. Second, as pointed out in Nijkamp et al. (2019a), the energy potentials learned with non-convergent MCMC do not have a valid steady-state, in the sense that samples from long-run Markov chains can differ greatly from observed samples, making it difficult to evaluate the learned energy potentials.

Another line of work, originating from Sohl-Dickstein et al. (2015), is to learn from a diffused version of the data, which are obtained from the original data via a diffusion process that sequentially adds Gaussian white noise. From such diffusion data, one can learn the conditional model of the data at a certain noise level given their noisy versions at the higher noise level of the diffusion process. After learning the sequence of conditional models that invert the diffusion process, one can then generate synthesized images from Gaussian white noise images by ancestral sampling. Building on

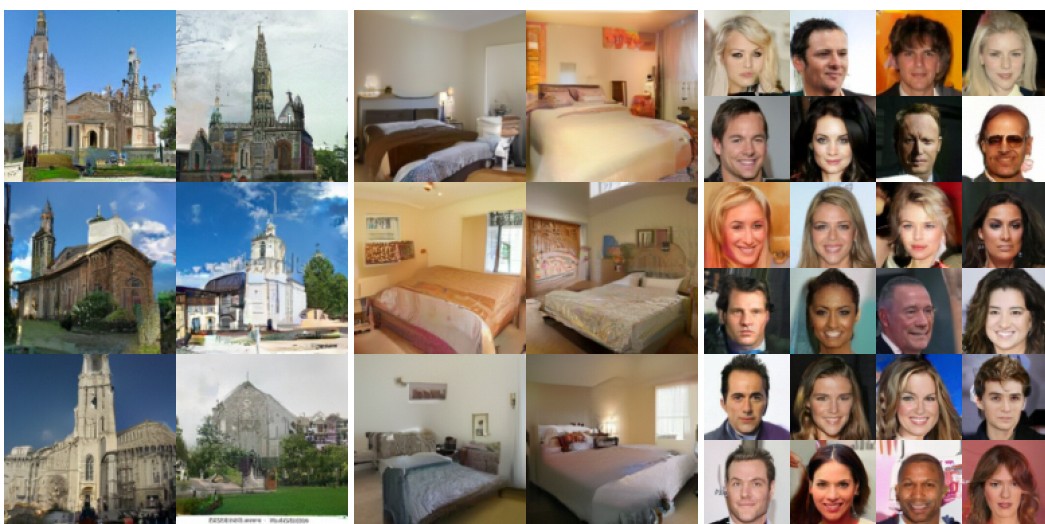

Figure 1: Generated samples on LSUN $128^2$ church_outdoor (*left*), LSUN $128^2$ bedroom (*center*) and CelebA $64^2$ (*right*).

Sohl-Dickstein et al. (2015), Ho et al. (2020) further developed the method, obtaining strong image synthesis results.

Inspired by Sohl-Dickstein et al. (2015) and Ho et al. (2020), we propose a *diffusion recovery likelihood* method to tackle the challenge of training EBMs directly on a dataset by instead learning a sequence of EBMs for the *marginal* distributions of the diffusion process. The sequence of marginal EBMs are learned with recovery likelihoods that are defined as the conditional distributions that invert the diffusion process. Compared to standard maximum likelihood estimation (MLE) of EBMs, learning marginal EBMs by diffusion recovery likelihood only requires sampling from the conditional distributions, which is much easier than sampling from the marginal distributions. After learning the marginal EBMs, we can generate synthesized images by a sequence of conditional samples initialized from the Gaussian white noise distribution. Unlike Ho et al. (2020) that approximates the reverse process by normal distributions, in our case the conditional distributions are derived from the marginal EBMs, which are more flexible. The framework of recovery likelihood was originally proposed in Bengio et al. (2013). In our work, we adapt it to learning the sequence of marginal EBMs from the diffusion data.

Our work is also related to the denoising score matching method of Vincent (2011), which was further developed by Song & Ermon (2019; 2020) for learning from diffusion data. The training objective used for diffusion probabilisitic models is a weighted version of the denoising score matching objective, as revealed by Ho et al. (2020). These methods learn the score functions (the gradients of the energy functions) directly, instead of using the gradients of learned energy functions as in EBMs. On the other hand, Saremi et al. (2018) parametrizes the score function as the gradient of a MLP energy function, and Saremi & Hyvarinen (2019) further unifies denoising score matching and neural empirical Bayes.

We demonstrate the efficacy of diffusion recovery likelihood on CIFAR-10, CelebA and LSUN datasets. The generated samples are of high fidelity and comparable to GAN-based methods. On CIFAR-10, we achieve FID 9.58 and inception score 8.30, exceeding existing methods of learning explicit EBMs to a large extent. We also demonstrate that diffusion recovery likelihood outperforms denoising score matching from diffusion data if we naively take the gradients of explicit energy functions as the score functions. More interestingly, by using a thousand diffusion time steps, we demonstrate that even very long MCMC chains from the sequence of conditional distributions produce samples that represent realistic images. With the faithful long-run MCMC samples from the conditional distributions, we can accurately estimate the marginal partition function at zero noise level by importance sampling, and thus evaluate the normalized density of data under the EBM.

Figure 3: Illustration of diffusion recovery likelihood on 2D checkerboard example. *Top*: progressively generated samples. *Bottom*: estimated marginal densities.

## 2 BACKGROUND

Let $\mathbf{x} \sim p_{\text{data}}(\mathbf{x})$ denote a training example, and $p_\theta(\mathbf{x})$ denote a model's probability density function that aims to approximates $p_{\text{data}}(\mathbf{x})$. An energy-based model (EBM) is defined as:

$$p_\theta(\mathbf{x}) = \frac{1}{Z_\theta} \exp(f_\theta(\mathbf{x})), \tag{1}$$

where $Z_\theta = \int \exp(f_\theta(\mathbf{x}))d\mathbf{x}$ is the partition function, which is analytically intractable for high-dimensional $\mathbf{x}$. For images, we parameterize $f_\theta(\mathbf{x})$ with a convolutional neural network with a scalar output.

The energy-based model in equation 1 can, in principle, be learned through MLE. Specifically, suppose we observe samples $\mathbf{x}_i \sim p_{\text{data}}(\mathbf{x})$ for $i = 1, 2, ..., n$. The log-likelihood function is

$$\mathcal{L}(\theta) = \frac{1}{n} \sum_{i=1}^{n} \log p_\theta(\mathbf{x}_i) \doteq \mathbb{E}_{\mathbf{x} \sim p_{\text{data}}}[\log p_\theta(\mathbf{x})]. \tag{2}$$

In MLE, we seek to maximize the log-likelihood function, where the gradient approximately follows (Xie et al., 2016b)

$$-\frac{\partial}{\partial \theta} D_{\text{KL}}(p_{\text{data}} \| p_\theta) = \mathbb{E}_{\mathbf{x} \sim p_{\text{data}}} \left[ \frac{\partial}{\partial \theta} f_\theta(\mathbf{x}) \right] - \mathbb{E}_{\mathbf{x} \sim p_\theta} \left[ \frac{\partial}{\partial \theta} f_\theta(\mathbf{x}) \right]. \tag{3}$$

The expectations can be approximated by averaging over the observed samples and the synthesized samples drawn from the model distribution $p_\theta(\mathbf{x})$ respectively. Generating synthesized samples from $p_\theta(\mathbf{x})$ can be done with Markov Chain Monte Carlo (MCMC) such as Langevin dynamics (or Hamiltonian Monte Carlo (Girolami & Calderhead, 2011)), which iterates

$$\mathbf{x}^{\tau+1} = \mathbf{x}^\tau + \frac{\delta^2}{2} \nabla_\mathbf{x} f_\theta(\mathbf{x}^\tau) + \delta \boldsymbol{\epsilon}^\tau, \tag{4}$$

where $\tau$ indexes the time, $\delta$ is the step size, and $\boldsymbol{\epsilon}^\tau \sim \mathcal{N}(0, \boldsymbol{I})$. The difficulty lies in the fact that for high-dimensional and multi-modal distributions, MCMC sampling can take a long time to converge, and the sampling chains may have difficulty traversing modes. As demonstrated in Figure 2, training EBMs with synthesized samples from non-convergent MCMC results in malformed energy landscapes (Nijkamp et al., 2019b), even if the samples from the model look reasonable.

Figure 2: Comparison of learning EBMs by diffusion recovery likelihood (Ours) versus marginal likelihood (Short-run).

## 3 RECOVERY LIKELIHOOD

### 3.1 FROM MARGINAL TO CONDITIONAL

Given the difficulty of sampling from the marginal density $p_\theta(\mathbf{x})$, following Bengio et al. (2013), we use the recovery likelihood defined by the density of the

observed sample conditional on a noisy sample perturbed by isotropic Gaussian noise. Specifically, let $\tilde{\mathbf{x}} = \mathbf{x} + \sigma\boldsymbol{\epsilon}$ be the noisy observation of $\mathbf{x}$, where $\boldsymbol{\epsilon} \sim \mathcal{N}(0, \boldsymbol{I})$. Suppose $p_\theta(\mathbf{x})$ is defined by the EBM in equation 1, then the conditional EBM can be derived as

$$p_\theta(\mathbf{x}|\tilde{\mathbf{x}}) = \frac{1}{\tilde{Z}_\theta(\tilde{\mathbf{x}})} \exp\left(f_\theta(\mathbf{x}) - \frac{1}{2\sigma^2}\|\tilde{\mathbf{x}} - \mathbf{x}\|^2\right), \tag{5}$$

where $\tilde{Z}_\theta(\tilde{\mathbf{x}}) = \int \exp\left(f_\theta(\mathbf{x}) - \frac{1}{2\sigma^2}\|\tilde{\mathbf{x}} - \mathbf{x}\|^2\right) d\mathbf{x}$ is the partition function of this conditional EBM. See Appendix A.1 for the derivation. Compared to $p_\theta(\mathbf{x})$ (equation 1), the extra quadratic term $\frac{1}{2\sigma^2}\|\tilde{\mathbf{x}} - \mathbf{x}\|^2$ in $p_\theta(\mathbf{x}|\tilde{\mathbf{x}})$ constrains the energy landscape to be localized around $\tilde{\mathbf{x}}$, making the latter less multi-modal and easier to sample from. As we will show later, when $\sigma$ is small, $p_\theta(\mathbf{x}|\tilde{\mathbf{x}})$ is approximately a single mode Gaussian distribution, which greatly reduces the burden of MCMC.

A more general formulation is $\tilde{\mathbf{x}} = a\mathbf{x} + \sigma\boldsymbol{\epsilon}$, where $a$ is a positive constant. In that case, we can let $\mathbf{y} = a\mathbf{x}$, and treat $\mathbf{y}$ as the observed sample. Assume $p_\theta(\mathbf{y}) = \frac{1}{Z_\theta}\exp(f_\theta(\mathbf{y}))$, then by *change of variable*, the density function of $\mathbf{x}$ can be derived as $g_\theta(\mathbf{x}) = a p_\theta(a\mathbf{x})$.

## 3.2 Maximizing Recovery Likelihood

With the conditional EBM, assume we have observed samples $\mathbf{x}_i \sim p_{\text{data}}(\mathbf{x})$ and the corresponding perturbed samples $\tilde{\mathbf{x}}_i = \mathbf{x}_i + \sigma\boldsymbol{\epsilon}_i$ for $i = 1, ..., n$. We define the *recovery log-likelihood function* as

$$\mathcal{J}(\theta) = \frac{1}{n}\sum_{i=1}^{n} \log p_\theta(\mathbf{x}_i|\tilde{\mathbf{x}}_i). \tag{6}$$

The term *recovery* indicates that we attempt to recover the clean sample $\mathbf{x}_i$ from the noisy sample $\tilde{\mathbf{x}}_i$. Thus, instead of maximizing $\mathcal{L}(\theta)$ in equation 2, we can maximize $\mathcal{J}(\theta)$, whose distributions are easier to sample from. Specifically, we generate synthesized samples by $K$ steps of Langevin dynamics that iterates

$$\mathbf{x}^{\tau+1} = \mathbf{x}^\tau + \frac{\delta^2}{2}\left(\nabla_\mathbf{x} f_\theta(\mathbf{x}^\tau) + \frac{1}{\sigma^2}(\tilde{\mathbf{x}} - \mathbf{x}^\tau)\right) + \delta\boldsymbol{\epsilon}^\tau. \tag{7}$$

The model is then updated following the same learning gradients as MLE (equation 3), because the quadratic term $-\frac{1}{2\sigma^2}\|\tilde{\mathbf{x}} - \mathbf{x}\|^2$ is not related to $\theta$. Following the classical analysis of MLE, we can show that the point estimate given by maximizing recovery likelihood is an unbiased estimator of the true parameters, which means that given enough data, a rich enough model and exact synthesis, maximizing the recovery likelihood learns $\theta$ such that $p_{\text{data}}(\mathbf{x}) = p_\theta(\mathbf{x})$. See Appendix A.2 for a theoretical explanation.

## 3.3 Normal Approximation to Conditional

When the variance of perturbed noise $\sigma^2$ is small, $p_\theta(\mathbf{x}|\tilde{\mathbf{x}})$ can be approximated by a normal distribution via a first order Taylor expansion at $\tilde{\mathbf{x}}$. Specifically, the negative conditional energy is

$$-\mathcal{E}_\theta(\mathbf{x}|\tilde{\mathbf{x}}) = f_\theta(\mathbf{x}) - \frac{1}{2\sigma^2}\|\tilde{\mathbf{x}} - \mathbf{x}\|^2 \tag{8}$$

$$\doteq f_\theta(\tilde{\mathbf{x}}) + \langle\nabla_\mathbf{x} f_\theta(\tilde{\mathbf{x}}), \mathbf{x} - \tilde{\mathbf{x}}\rangle - \frac{1}{2\sigma^2}\|\tilde{\mathbf{x}} - \mathbf{x}\|^2 \tag{9}$$

$$= -\frac{1}{2\sigma^2}\left[\|\mathbf{x} - (\tilde{\mathbf{x}} + \sigma^2\nabla_\mathbf{x} f_\theta(\tilde{\mathbf{x}}))\|^2\right] + c, \tag{10}$$

where $c$ include terms irrelevant of $\mathbf{x}$ (see Appendix A.3 for a detailed derivation). In the above approximation, we do not perform second order Taylor expansion because $\sigma^2$ is small, and $\|\tilde{\mathbf{x}} - \mathbf{x}\|^2/2\sigma^2$ will dominate all the second order terms from Taylor expansion. Thus we can approximate $p_\theta(\mathbf{x}|\tilde{\mathbf{x}})$ by a Gaussian approximation $\widetilde{p}_\theta(\mathbf{x}|\tilde{\mathbf{x}})$:

$$\widetilde{p}_\theta(\mathbf{x}|\tilde{\mathbf{x}}) = \mathcal{N}\left(\mathbf{x}; \tilde{\mathbf{x}} + \sigma^2\nabla_\mathbf{x} f_\theta(\tilde{\mathbf{x}}), \sigma^2\right). \tag{11}$$

We can sample from this distribution using:

$$\mathbf{x}_{\text{gen}} = \tilde{\mathbf{x}} + \sigma^2\nabla_\mathbf{x} f_\theta(\tilde{\mathbf{x}}) + \sigma\boldsymbol{\epsilon}, \tag{12}$$

where $\boldsymbol{\epsilon} \sim \mathcal{N}(0, I)$. This resembles a single step of Langevin dynamics, except that $\sigma\boldsymbol{\epsilon}$ is replaced by $\sqrt{2}\sigma\boldsymbol{\epsilon}$ in Langevin dynamics. This normal approximation has two traits: (1) it verifies the fact that the conditional density $p_\theta(\mathbf{x}|\tilde{\mathbf{x}})$ can be generally easier to sample from when $\sigma$ is small; (2) it provides hints of choosing the step size of Langevin dynamics, as discussed in section 3.5.

### 3.4 CONNECTION TO VARIATIONAL INFERENCE AND SCORE MATCHING

The normal approximation to the conditional distribution leads to a natural connection to diffusion probabilistic models (Sohl-Dickstein et al., 2015; Ho et al., 2020) and denoising score matching (Vincent, 2011; Song & Ermon, 2019; 2020; Saremi et al., 2018; Saremi & Hyvarinen, 2019). Specifically, in diffusion probabilistic models, instead of modeling $p_\theta(x)$ as an energy-based model, it recruits variational inference and directly models the conditional density as

$$p_\theta(\mathbf{x}|\tilde{\mathbf{x}}) \sim \mathcal{N}\left(\tilde{\mathbf{x}} + \sigma^2 s_\theta(\tilde{\mathbf{x}}), \sigma^2\right), \tag{13}$$

which is in agreement with the normal approximation (equation 11), with $s_\theta(\mathbf{x}) = \nabla_{\mathbf{x}} f_\theta(\mathbf{x})$. On the other hand, the training objective of denoising score matching is to minimize

$$\frac{1}{2\sigma^2}\mathbb{E}_{p(\mathbf{x},\tilde{\mathbf{x}})}[\|\mathbf{x} - (\tilde{\mathbf{x}} + \sigma^2 s_\theta(\tilde{\mathbf{x}}))\|^2], \tag{14}$$

where $s_\theta(\mathbf{x})$ is the score of the density of $\tilde{\mathbf{x}}$. This objective is in agreement with the objective of maximizing log-likelihood of the normal approximation (equation 10), except that for normal approximation, $\nabla_{\mathbf{x}} f_\theta(\cdot)$ is the score of density of $\mathbf{x}$, instead of $\tilde{\mathbf{x}}$. However, the difference between the scores of density of $\mathbf{x}$ and $\tilde{\mathbf{x}}$ is of $O(\sigma^2)$, which is negligible when $\sigma$ is sufficiently small (see Appendix A.4 for details). We can further show that the learning gradient of maximizing log-likelihood of the normal approximation is approximately the same as the learning gradient of maximizing the original recovery log-likelihood with one step of Langevin dynamics (see Appendix A.5). It indicates that the training process of maximizing recovery likelihood agrees with the one of diffusion probabilistic models and denoising score matching when $\sigma$ is small.

As the normal approximation is accurate only when $\sigma$ is small, it requires many time steps in the diffusion process for this approximation to work well, which is also reported in Ho et al. (2020) and Song & Ermon (2020). In contrast, the diffusion recovery likelihood framework can be more flexible in choosing the number of time steps and the magnitude of $\sigma$.

### 3.5 DIFFUSION RECOVERY LIKELIHOOD

As we discuss, sampling from $p_\theta(\mathbf{x}|\tilde{\mathbf{x}})$ becomes simple only when $\sigma$ is small. In the extreme case when $\sigma \to \infty$, $p_\theta(\mathbf{x}|\tilde{\mathbf{x}})$ converges to the marginal distribution $p_\theta(\mathbf{x})$, which is again highly multimodal and difficult to sample from. To keep $\sigma$ small and meanwhile equip the model with the ability to generate new samples initialized from white noise, inspired by Sohl-Dickstein et al. (2015) and Ho et al. (2020), we propose to learn a sequence of recovery likelihoods, on gradually perturbed observed data based on a diffusion process. Specifically, assume a sequence of perturbed observations $\mathbf{x}_0, \mathbf{x}_1, ..., \mathbf{x}_T$ such that

$$\mathbf{x}_0 \sim p_{\text{data}}(\mathbf{x}); \quad \mathbf{x}_{t+1} = \sqrt{1 - \sigma_{t+1}^2}\mathbf{x}_t + \sigma_{t+1}\boldsymbol{\epsilon}_{t+1}, \quad t = 0, 1, ...T - 1. \tag{15}$$

The scaling factor $\sqrt{1 - \sigma_{t+1}^2}$ ensures that the sequence is a spherical interpolation between the observed sample and Gaussian white noise. Let $\mathbf{y}_t = \sqrt{1 - \sigma_{t+1}^2}\mathbf{x}_t$, and we assume a sequence of conditional EBMs

$$p_\theta(\mathbf{y}_t|\mathbf{x}_{t+1}) = \frac{1}{\tilde{Z}_{\theta,t}(\mathbf{x}_{t+1})}\exp\left(f_\theta(\mathbf{y}_t, t) - \frac{1}{2\sigma_{t+1}^2}\|\mathbf{x}_{t+1} - \mathbf{y}_t\|^2\right), \quad t = 0, 1, ..., T - 1, \tag{16}$$

where $f_\theta(\mathbf{y}_t, t)$ is defined by a neural network conditioned on $t$.

We follow the learning algorithm in section 3.2. A question is how to determine the step size schedule $\delta_t$ of Langevin dynamics. Inspired by the sampling procedure of the normal approximation (equation 12), we set the step size $\delta_t = b\sigma_t$, where $b < 1$ is a tuned hyperparameter. This schedule turns out to work well in practice. Thus the $K$ steps of Langevin dynamics iterates

$$\mathbf{y}_t^{\tau+1} = \mathbf{y}_t^\tau + \frac{b^2\sigma_t^2}{2}(\nabla_{\mathbf{y}} f_\theta(\mathbf{y}_t^\tau, t) + \frac{1}{\sigma_t^2}(\mathbf{x}_{t+1} - \mathbf{y}_t^\tau)) + b\sigma_t\boldsymbol{\epsilon}^\tau. \tag{17}$$

Algorithm 1 summarizes the training procedure. After training, we initialize the MCMC sampling from Gaussian white noise, and the synthesized sample at each time step serves to initialize the MCMC that samples from the model of the previous time step. See algorithm 2. To show the efficacy of our method, Figures 3 and 2 display several 2D toy examples learned by diffusion recovery likelihood.

---

**Algorithm 1** Training

**repeat**
    Sample $t \sim \mathrm{Unif}(\{0, ..., T-1\})$.
    Sample pairs $(\mathbf{y}_t, \mathbf{x}_{t+1})$.
    Set synthesized sample $\mathbf{y}_t^- = \mathbf{x}_{t+1}$.
    **for** $\tau \leftarrow 1$ to $K$ **do**
        Update $\mathbf{y}_t^-$ according to equation 17.
    **end for**
    Update $\theta$ following the gradients
    $\frac{\partial}{\partial \theta} f_\theta(\mathbf{y}_t, t) - \frac{\partial}{\partial \theta} f_\theta(\mathbf{y}_t^-, t)$.
**until** converged.

---

**Algorithm 2** Progressive sampling

Sample $\mathbf{x}_T \sim \mathcal{N}(0, \boldsymbol{I})$.
**for** $t \leftarrow T-1$ to $0$ **do**
    $\mathbf{y}_t = \mathbf{x}_{t+1}$.
    **for** $\tau \leftarrow 1$ to $K$ **do**
        Update $\mathbf{y}_t$ according to equation 17.
    **end for**
    $\mathbf{x}_t = \mathbf{y}_t / \sqrt{1 - \sigma_{t+1}^2}$.
**end for**
**return** $\mathbf{x}_0$.

---

## 4 EXPERIMENTS

To show that diffusion recovery likelihood is flexible for diffusion process of various magnitudes of noise, we test the method under two settings: (1) $T = 6$, with $K = 30$ steps of Langevin dynamic per time step; (2) $T = 1000$, with sampling from normal approximation. (2) resembles the noise schedule of Ho et al. (2020) and the magnitude of noise added at each time step is much smaller compared to (1). For both settings, we set $\sigma_t^2$ to increase linearly. The network structure of $f_\theta(x, t)$ is based on Wide ResNet (Zagoruyko & Komodakis, 2016) and we remove weight normalization. $t$ is encoded by Transformer sinusoidal position embedding as in (Ho et al., 2020). For (1), we find that adding another scaling factor $c_t$ to the step size $\delta_t$ helps. Architecture and training details are in Appendix B. Henceforth we simply refer the two settings as *T6* and *T1k*.

### 4.1 IMAGE GENERATION

Figures 1 and 4 display uncurated samples generated from learned models on CIFAR-10, CelebA $64 \times 64$, LSUN $64 \times 64$ and $128 \times 128$ datasets under *T6* setting. The samples are of high fidelity and comparable to GAN-based methods. Appendix C.5 provides more generated samples. Tables 1 and 3 summarize the quantitative evaluations on CIFAR-10 and CelebA datasets, in terms of Frechet Inception Distance (FID) (Heusel et al., 2017) and inception scores (Salimans et al., 2016). On CIFAR-10, our model achieves FID 9.58 and inception score 8.30, which outperforms existing methods of learning explicit energy-based models to a large extent, and is superior to a majority of GAN-based methods. On CelebA, our model obtains results comparable with the state-of-the-art GAN-based methods, and outperforms score-based methods (Song & Ermon, 2019; 2020). Note that the score-based methods (Song & Ermon, 2019; 2020) and diffusion probabilistic models (Ho et al., 2020) directly parametrize and learn the score of data distribution, whereas our goal is to learn explicit energy-based models.

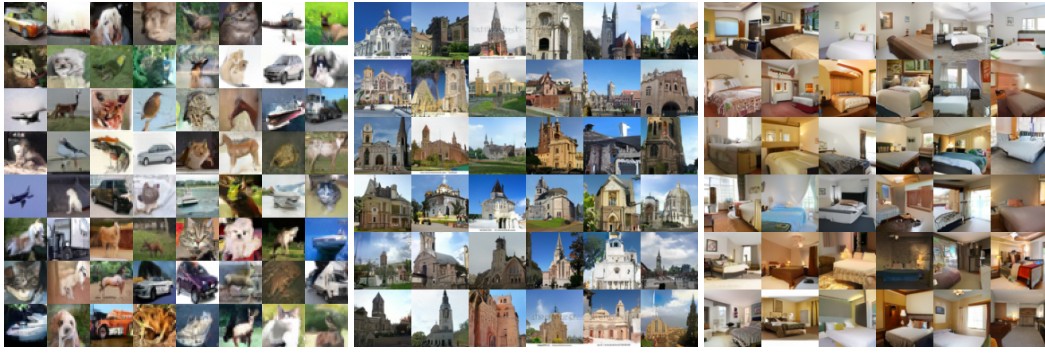

Figure 4: Generated samples on unconditional CIFAR-10 (*left*) and LSUN $64^2$ church_outdoor (*center*) and LSUN $64^2$ bedroom (*right*).

Table 1: FID and inception scores on CIFAR-10.

| Model | FID↓ | Inception↑ |
|---|---|---|
| **GAN-based** | | |
| WGAN-GP (Gulrajani et al., 2017) | 36.4 | $7.86 \pm .07$ |
| SNGAN (Miyato et al., 2018) | 21.7 | $8.22 \pm .05$ |
| SNGAN-DDLS (Che et al., 2020) | 15.42 | $9.09 \pm .10$ |
| StyleGAN2-ADA (Karras et al., 2020) | 3.26 | $\mathbf{9.74 \pm .05}$ |
| **Score-based** | | |
| NCSN (Song & Ermon, 2019) | 25.32 | $8.87 \pm .12$ |
| NCSN-v2 (Song & Ermon, 2020) | 10.87 | $8.40 \pm .07$ |
| DDPM (Ho et al., 2020) | **3.17** | $9.46 \pm .11$ |
| **Explicit EBM-conditional** | | |
| CoopNets (Xie et al., 2019) | - | 7.30 |
| EBM-IG (Du & Mordatch, 2019) | 37.9 | 8.30 |
| JEM (Grathwohl et al., 2019) | 38.4 | 8.76 |
| **Explicit EBM** | | |
| Muli-grid (Gao et al., 2018) | 40.01 | 6.56 |
| CoopNets (Xie et al., 2016a) | 33.61 | 6.55 |
| EBM-SR (Nijkamp et al., 2019b) | - | 6.21 |
| EBM-IG (Du & Mordatch, 2019) | 38.2 | 6.78 |
| **Ours** (*T6*) | **9.58** | $\mathbf{8.30} \pm .11$ |

Table 2: Ablation of training objectives, time steps $T$ and sampling steps $K$ on CIFAR-10. $K = 0$ indicates that we sample from the normal approximation.

| Setting / Objective | FID↓ | Inception↑ |
|---|---|---|
| T = 1, K = 180 | 32.12 | $6.72 \pm 0.12$ |
| T = 1000, K = 0 | 22.58 | $7.71 \pm 0.08$ |
| T = 1000, K = 0 (DSM) | 21.76 | $7.76 \pm 0.11$ |
| T = 6, K = 10 | - | - |
| T = 6, K = 30 | 9.58 | $8.30 \pm 0.11$ |
| T = 6, K = 50 | **9.36** | $\mathbf{8.46 \pm 0.13}$ |

Table 3: FID scores on CelebA $64^2$.

| Model | FID↓ |
|---|---|
| QA-GAN (Parimala & Channappayya, 2019) | 6.42 |
| COCO-GAN (Lin et al., 2019) | **4.0** |
| NVAE (Vahdat & Kautz, 2020) | 14.74 |
| NCSN (Song & Ermon, 2019) | 25.30 |
| NCSN-v2 (Song & Ermon, 2020) | 10.23 |
| EBM-SR (Nijkamp et al., 2019b) | 23.02 |
| EBM-Triangle (Han et al., 2020) | 24.70 |
| **Ours** (*T6*) | 5.98 |

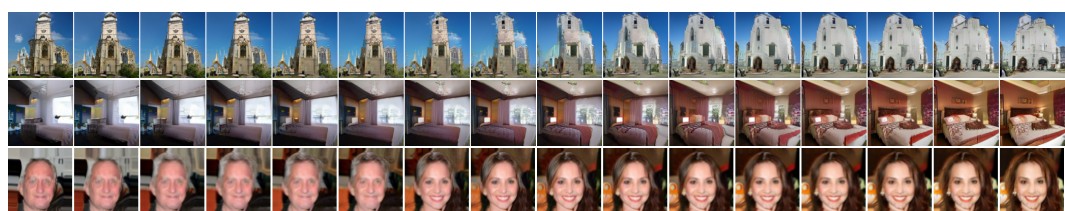

Figure 5: Interpolation results between the leftmost and rightmost generated samples. For *top* to *bottom*: LSUN church_outdoor $128^2$, LSUN bedroom $128^2$ and CelebA $64^2$.

Table 4: Test bits per dimension on CIFAR-10. [†] indicates that we estimate the bit per dimension with the approximated log partition function instead of analytically computing it. See section 4.2.

| Model | BPD↓ |
|---|---|
| DDPM (Ho et al., 2020) | 3.70 |
| Glow (Kingma & Dhariwal, 2018) | 3.35 |
| Flow++ (Ho et al., 2019) | 3.08 |
| GPixelCNN (Van den Oord et al., 2016) | 3.03 |
| Sparse Transformer (Child et al., 2019) | 2.80 |
| DistAug (Jun et al., 2020) | **2.56** |
| **Ours**[†] (*T1k*) | 3.18 |

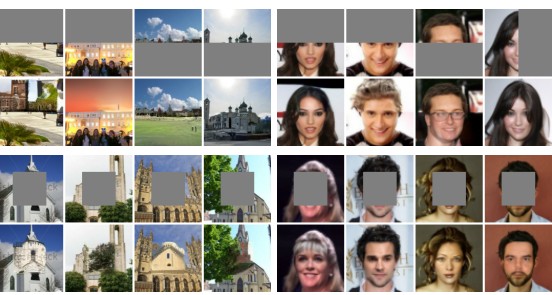

Figure 6: Image inpainting on LSUN church_outdoor $128^2$ (*left*) and CelebA $64^2$ (*right*). With each block, the top row are mask images while the bottom row are inpainted images.

**Interpolation.** As shown in Figure 5, our model is capable of smooth interpolation between two generated samples. Specifically, for two samples $\mathbf{x}_0^{(0)}$ and $\mathbf{x}_0^{(1)}$, we do a sphere interpolation between

the initial white noise images $\mathbf{x}_T^{(0)}$ and $\mathbf{x}_T^{(1)}$ and the noise terms of Langevin dynamics $\boldsymbol{\epsilon}_{t,\tau}^{(0)}$ and $\boldsymbol{\epsilon}_{t,\tau}^{(1)}$ for every sampling step at every time step. More interpolation results can be found in Appendix C.3.

**Image inpainting.** A promising application of energy-based models is to use the learned model as a prior model for image processing, such as image inpainting, denoising and super-resolution (Gao et al., 2018; Du & Mordatch, 2019; Song & Ermon, 2019). In Figure 6, we demonstrate that the learned models by maximizing recovery likelihoods are capable of realistic and semantically meaningful image inpainting. Specifically, given a masked image and the corresponding mask, we first obtain a sequence of perturbed masked images at different noise levels. The inpainting can be easily achieved by running Langevin dynamics progressively on the masked pixels while keeping the observed pixels fixed at decreasingly lower noise levels. Additional image inpainting results can be found in Appendix C.4.

**Ablation study.** Table 2 summarizes the results of ablation study on CIFAR-10. We investigate the effect of changing the numbers of time steps $T$ and sampling steps $K$. First, to show that it is beneficial to learn by diffusion recovery likelihood, we compare against a baseline approach ($T = 1, K = 180$) where we use only one time step, so that the recovery likelihood becomes marginal likelihood. The approach is adopted by Nijkamp et al. (2019b) and Du & Mordatch (2019). For fair comparison, we equip the baseline method the same budget of MCMC sampling as our *T6* setting (i.e., 180 sampling steps). Our method outperforms this baseline method by a large margin. Also the models are trained more efficiently as the number of sampling steps per iteration is reduced and amortized by time steps.

Next, we report the sample quality of setting *T1k*. We test two training objectives for this setting: (1) maximizing recovery likelihoods (T = 1000, K = 0) and (2) maximizing the approximated normal distributions (T=1000, K=0 (DSM)). As mentioned in section 3.4, (2) is equivalent to the training objectives of denoising score matching (Song & Ermon, 2019; 2020) and diffusion probabilistic model (Ho et al., 2020), except that the score functions are taken as the gradients of explicit energy functions. In practice, for a direct comparison, (2) follows the same implementation as in Ho et al. (2020), except that the score function is parametrized as the gradients of the explicit energy function used in our method. (1) and (2) achieve similar sample quality in terms of quantitative metrics, where (2) results in a slightly better FID score yet a slightly worse inception score. This verifies the fact that the training objectives of (1) and (2) are consistent. Both (1) and (2) performs worse than setting T6. A possible explanation is that the sampling error may accumulate with many time steps, so that a more flexible schedule of time steps accompanied with certain amount of sampling steps is preferred.

Last, we examine the influence of varying the number of sampling steps while fixing the number of time steps. The training becomes unstable when the number of sampling steps are not enough ($T = 6, K = 10$), and more sampling steps lead to better sample quality. However, since $K = 50$ does not gain significant improvement versus $K = 30$, yet of much higher computational cost, we keep $K = 30$ for image generation on all datasets. See Appendix C.1 for a plot of FID scores over iterations.

## 4.2 LONG-RUN CHAIN ANALYSIS

Besides achieving high quality generation, a perhaps equally important aspect of learning EBMs is to obtain a faithful energy potential. A principle way to check the validity of the learned potential is to perform long-run sampling chains and see if the samples still remain realistic. However, as pointed out in Nijkamp et al. (2019a), almost all existing methods of learning EBMs fail in getting realistic long-run chain samples. In this subsection, we demonstrate that by composing a thousand diffusion time steps (*T1k* setting), we can form steady long-run MCMC chains for the conditional distributions.

First we prepare a faithful sampler for conducting long-run sampling. Specifically, after training the model under $T1k$ setting by maximizing diffusion recovery likelihood, for each time step, we first sample from the normal approximation and count it as one sampling step, and then use Hamiltonian Monte Carlo (HMC) (Neal et al., 2011) with 2 leapfrog steps to perform the consecutive sampling steps. To obtain a reasonable schedule of sampling step size, for each time step we adaptively adjust the step size of HMC to make the average acceptance rate range in $[0.6, 0.9]$, which is computed

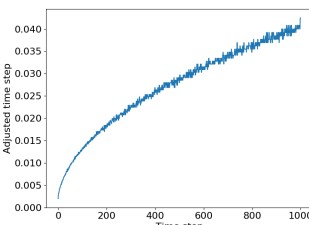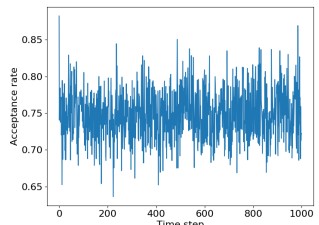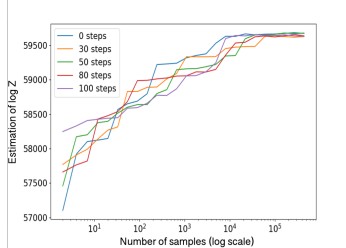

Figure 7: *Left*: Adjusted step size of HMC over time step. *Center*: Acceptance rate over time step. *Right*: Estimated log partition function over number of samples with different number of sampling steps per time step. The x axis is plotted in log scale.

over 1000 chains for 100 steps. Figure 7 displays the adjusted step size (*left*) and acceptance rate (*center*) over time step. The adjusted step size increases logarithmically. With this step size schedule, we generate long-run chains from the learned sequence of conditional distributions. As shown in Figure 8, images remain realistic for even $100k$ sampling steps in total (i.e., 100 sampling steps per time step), resulting in FID 24.89. This score is close to the one computed on samples generated by $1k$ steps (i.e., sampled from normal approximation), which is 25.12. As a further check, we recruit a No-U-Turn Sampler (Hoffman & Gelman, 2014) with the same step size schedule as HMC to perform long-run sampling, where the samples also remain realistic. See Appendix C.2 for details.

More interestingly, given the faithful long-run MCMC samples from the *conditional* distributions, we can estimate the log ratio of the partition functions of the *marginal* distributions, and further estimate the partition function of $p_\theta(\mathbf{y}_0)$. The strategy is based on annealed importance sampling (Neal, 2001). See Appendix A.6 for the implementation details. The right subfigure of Figure 7 depicts the estimated log partition function of $p_\theta(\mathbf{y}_0)$ over the number of MCMC samples used. To verify the estimation strategy and again check the long-run chain samples, we conduct multiple runs using samples generated with different numbers of HMC steps and display the estimation curves. All the

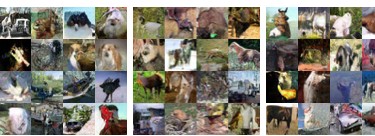

Figure 8: Long-run chain samples from model-*T1k* with different total amount of HMC steps. From *left* to *right*: $1k$ steps, $10k$ steps and $100k$ steps.

curves saturate to values close to each other at the end, indicating the stability of long-run chain samples and the effectiveness of the estimation strategy. With the estimated partition function, by *change of variable*, we can estimate the normalized density of data as $g_\theta(\mathbf{x}_0) = \sqrt{1 - \sigma_1^2} p_\theta(\sqrt{1 - \sigma_1^2}\mathbf{x}_0)$. We report test bits per dimension on CIFAR-10 in Table 4. Note that the result should be taken with a grain of salt, because the partition function is estimated by samples and as shown in Appendix A.6, it is a stochastic lower bound of the true value, that will converge to the true value when the number of samples grows large.

## 5    CONCLUSION

We propose to learn EBMs by diffusion recovery likelihood, a variant of MLE applied to diffusion processes. We achieve high quality image synthesis, and with a thousand noise levels, we obtain faithful long-run MCMC samples that indicate the validity of the learned energy potentials. Since this method can learn EBMs efficiently with small budget of MCMC, we are also interested in scaling it up to higher resolution images and investigating this method in other data modalities in the future.

## ACKNOWLEDGEMENT

The work was done while Ruiqi Gao and Yang Song were interns at Google Brain during the summer of 2020. The work of Ying Nian Wu is supported by NSF DMS-2015577. We thank Alexander A. Alemi, Jonathan Ho, Tim Salimans and Kevin Murphy for their insightful discussions during the course of this project.

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

## A EXTENDED DERIVATIONS

### A.1 DERIVATION OF EQUATION 5

Let $\tilde{\mathbf{x}} = \mathbf{x} + \sigma\epsilon$, where $\epsilon \sim \mathcal{N}(0, \boldsymbol{I})$. Given the marginal distribution of

$$p_\theta(\mathbf{x}) = \frac{1}{Z_\theta}\exp(f_\theta(\mathbf{x})), \tag{18}$$

We can derive the conditional distribution of $\mathbf{x}$ given $\tilde{\mathbf{x}}$ as

$$p_\theta(\mathbf{x}|\tilde{\mathbf{x}}) = p_\theta(\mathbf{x})p(\tilde{\mathbf{x}}|\mathbf{x})/p(\tilde{\mathbf{x}}) \tag{19}$$

$$= \frac{1}{Z_\theta}\exp(f_\theta(\mathbf{x}))\frac{1}{(2\pi\sigma^2)^{\frac{n}{2}}}\exp(-\frac{1}{2\sigma^2}\|\tilde{\mathbf{x}} - \mathbf{x}\|^2)/p(\tilde{\mathbf{x}}) \tag{20}$$

$$= \frac{1}{\tilde{Z}_\theta(\tilde{\mathbf{x}})}\exp\left(f_\theta(\mathbf{x}) - \frac{1}{2\sigma^2}\|\tilde{\mathbf{x}} - \mathbf{x}\|^2\right), \tag{21}$$

where we absorb all the terms that are irrelevant of $\mathbf{x}$ as $\tilde{Z}_\theta(\tilde{\mathbf{x}})$.

### A.2 THEORETICAL UNDERSTANDING

In this subsection, we analyze the asymptotic behavior of maximizing the recovery log-likelihood.

For model class $\{p_\theta(\mathbf{x}), \forall\theta\}$, suppose there exists $\theta^*$ such that $p_{\text{data}} = p_{\theta^*}$. According to the classical theory of MLE, let $\hat{\theta}_0$ be the point estimate by MLE. Then we have $\hat{\theta}$ is an unbiased estimator of $\theta^*$ with asymptotic normality:

$$\sqrt{n}(\hat{\theta}_0 - \theta^*) \to \mathcal{N}(0, \mathcal{I}_0(\theta^*)^{-1}), \tag{22}$$

where $\mathcal{I}_0(\theta) = \mathbb{E}_{\mathbf{x}\sim p_\theta}[-\nabla_\theta^2 \log p_\theta(\mathbf{x})]$ is the Fisher information, and $n$ is the number of observed samples.

Let $\hat{\theta}$ be the point estimate given by maximizing recovery log-likelihood, we can derive a result in parallel to that of MLE:

$$\sqrt{n}(\hat{\theta} - \theta^*) \to \mathcal{N}(0, \mathcal{I}(\theta^*)^{-1}), \tag{23}$$

where $\mathcal{I}(\theta) = \mathbb{E}_{p_\theta(\mathbf{x},\tilde{\mathbf{x}})}[-\nabla_\theta^2 \log p_\theta(\mathbf{x}|\tilde{\mathbf{x}})]$. The relationship between $I_0(\theta)$ and $I(\theta)$ is that

$$\mathcal{I}_0(\theta) = \mathcal{I}(\theta) + \mathbb{E}_{p_\theta(\mathbf{x},\tilde{\mathbf{x}})}[-\nabla_\theta^2 \log p_\theta(\tilde{\mathbf{x}})]. \tag{24}$$

Thus there is loss of information, but $\hat{\theta}$ is still an unbiased estimator of $\theta^*$ with asymptotic normality.

### A.3 DETAILED DERIVATION OF NORMAL APPROXIMATION

$$-\mathcal{E}_\theta(\mathbf{x}|\tilde{\mathbf{x}}) = f_\theta(\mathbf{x}) - \frac{1}{2\sigma^2}\|\tilde{\mathbf{x}} - \mathbf{x}\|^2 \tag{25}$$

$$\doteq f_\theta(\tilde{\mathbf{x}}) + \langle\nabla_\mathbf{x}f_\theta(\tilde{\mathbf{x}}), \mathbf{x} - \tilde{\mathbf{x}}\rangle - \frac{1}{2\sigma^2}\|\tilde{\mathbf{x}} - \mathbf{x}\|^2 \tag{26}$$

$$= -\frac{1}{2\sigma^2}\left[\|\mathbf{x}\|^2 - 2\langle\tilde{\mathbf{x}}, \mathbf{x}\rangle + \|\tilde{\mathbf{x}}\|^2\right] + \langle\nabla_\mathbf{x}f_\theta(\tilde{\mathbf{x}}), \mathbf{x}\rangle - \langle\nabla_\mathbf{x}f_\theta(\tilde{\mathbf{x}}), \tilde{\mathbf{x}}\rangle + f_\theta(\tilde{\mathbf{x}}) \tag{27}$$

$$= -\frac{1}{2\sigma^2}\left[\|\mathbf{x}\|^2 - 2\langle\tilde{\mathbf{x}} + \sigma^2\nabla_\mathbf{x}f_\theta(\tilde{\mathbf{x}}), \mathbf{x}\rangle\right] - \frac{1}{2\sigma^2}\|\tilde{\mathbf{x}}\|^2 - \langle\nabla_\mathbf{x}f_\theta(\tilde{\mathbf{x}}), \tilde{\mathbf{x}}\rangle + f_\theta(\tilde{\mathbf{x}}) \tag{28}$$

$$= -\frac{1}{2\sigma^2}\left[\|\mathbf{x} - (\tilde{\mathbf{x}} + \sigma^2\nabla_\mathbf{x}f_\theta(\tilde{\mathbf{x}}))\|^2\right] + c, \tag{29}$$

### A.4 DIFFERENCE BETWEEN THE SCORES OF $p(\mathbf{x})$ AND $p(\tilde{\mathbf{x}})$

For notation clarity, with $\tilde{\mathbf{x}} = \mathbf{x} + \epsilon$, we let $\widetilde{p}$ be the distribution of $\tilde{\mathbf{x}}$, and $p$ be the distribution of $\mathbf{x}$. Then for a smooth testing function with vanishing tails,

$$\mathbb{E}[h(\tilde{\mathbf{x}})] = \mathbb{E}[h(\mathbf{x} + \epsilon)] \tag{30}$$

$$\doteq \mathbb{E}[h(\mathbf{x}) + h'(\mathbf{x})\epsilon + h''(\mathbf{x})\epsilon^2/2] \tag{31}$$

$$= \mathbb{E}[h(\mathbf{x})] + \mathbb{E}[h''(\mathbf{x})]\sigma^2/2. \tag{32}$$

Integral by parts,

$$\mathbb{E}[h''(\mathbf{x})] = \int h''(\mathbf{x})p(\mathbf{x})d\mathbf{x} = -\int h'(\mathbf{x})p'(\mathbf{x})d\mathbf{x} = \int p''(\mathbf{x})h(\mathbf{x})d\mathbf{x}. \tag{33}$$

Thus we have the heat equation

$$\widetilde{p}(\mathbf{x}) = p(\mathbf{x}) + p''(\mathbf{x})\sigma^2/2. \tag{34}$$

The score

$$\nabla_{\mathbf{x}} \log \tilde{p}(\mathbf{x}) = \nabla_x \log p(\mathbf{x}) + \nabla_{\mathbf{x}} \log(1 + p''(\mathbf{x})/p(\mathbf{x})\sigma^2/2) \tag{35}$$
$$\doteq \nabla_{\mathbf{x}} \log p(\mathbf{x}) + \nabla_{\mathbf{x}}[p''(\mathbf{x})/p(\mathbf{x})]\sigma^2/2. \tag{36}$$

Thus the difference between the score of $p$ and $\widetilde{p}$ is of the order $\sigma^2$, which is negligible when $\sigma^2$ is small.

### A.5 LEARNING GRADIENTS OF NORMAL APPROXIMATION AND ORIGINAL RECOVERY LIKELIHOOD

In this subsection we demonstrate that the learning gradient of maximizing likelihood of the normal approximation is approximately the same as the gradient of maximizing the original recovery likelihood with one step of Langevin sampling. Specifically, the gradient of the normal approximation of recovery log-likelihood for an observed $\mathbf{x}_{\text{obs}}$ is

$$\nabla_\theta \left( \frac{1}{2\sigma^2} \left[ \|\mathbf{x}_{\text{obs}} - (\tilde{\mathbf{x}} + \sigma^2 f'_\theta(\tilde{\mathbf{x}}))\|^2 \right] \right) = \nabla_\theta f'_\theta(\tilde{\mathbf{x}})(\mathbf{x}_{\text{obs}} - (\tilde{\mathbf{x}} + \sigma^2 f'_\theta(\tilde{\mathbf{x}})). \tag{37}$$

On the other hand, to maximize the original recovery likelihood, suppose we sample $\mathbf{x}_{\text{syn}} \sim p_\theta(\mathbf{x}|\tilde{\mathbf{x}})$, then the gradient ascent of the original recovery log-likelihood is

$$\nabla_\theta f_\theta(\mathbf{x}_{\text{obs}}) - \mathbb{E}[\nabla_\theta f_\theta(\mathbf{x}_{\text{syn}})] = h_\theta(\mathbf{x}_{\text{obs}}) - \mathbb{E}[h_\theta(\mathbf{x}_{\text{syn}})], \tag{38}$$

where $h_\theta(\mathbf{x}) = \nabla_\theta f_\theta(\mathbf{x})$. Approximately, if we perform one step of Langevin dynamics from $\tilde{\mathbf{x}}$ to obtain $\mathbf{x}_{\text{syn}}$, i.e., $x_{\text{syn}} = \tilde{\mathbf{x}} + \sigma^2 f'_\theta(\tilde{\mathbf{x}}) + \sqrt{2}\sigma e$, and assume $f_\theta(\mathbf{x})$ is locally linear in $\mathbf{x}$, then

$$\nabla_\theta f_\theta(\mathbf{x}_{\text{obs}}) - \mathbb{E}[\nabla_\theta f_\theta(\mathbf{x}_{\text{init}})] \tag{39}$$
$$= h_\theta(\mathbf{x}_{\text{obs}}) - \mathbb{E}[h_\theta(\tilde{\mathbf{x}} + \sigma^2 f'_\theta(\tilde{\mathbf{x}}) + \sigma e)] \tag{40}$$
$$\doteq h_\theta(\tilde{\mathbf{x}}) + h'_\theta(\tilde{\mathbf{x}})(x_{\text{obs}} - \tilde{\mathbf{x}}) - \mathbb{E}[h_\theta(\tilde{\mathbf{x}}) + h'_\theta(\tilde{\mathbf{x}})(\sigma^2 f'_\theta(\tilde{\mathbf{x}}) + \sigma e)] \tag{41}$$
$$= h'_\theta(\tilde{\mathbf{x}})(\mathbf{x}_{\text{obs}} - (\tilde{\mathbf{x}} + \sigma^2 f'_\theta(\tilde{\mathbf{x}})) \tag{42}$$
$$= \nabla_\theta f'_\theta(\tilde{\mathbf{x}})(\mathbf{x}_{\text{obs}} - (\tilde{\mathbf{x}} + \sigma^2 f'_\theta(\tilde{\mathbf{x}})). \tag{43}$$

Comparing equations 37 and 43, we see that the two gradients agree with each other.

### A.6 ESTIMATING THE PARTITION FUNCTION

We can utilize the sequence of learned distributions of $\mathbf{y}_t$ $(= \sqrt{1 - \sigma_{t+1}^2}\mathbf{x}_t)$ to estimate the partition function. Specifically, the marginal distribution of $\mathbf{y}_t$ is

$$p_\theta(\mathbf{y}_t) = \frac{1}{Z_{\theta,t}} \exp\left(f_\theta(\mathbf{y}_t, t)\right) \tag{44}$$

We can estimate the ratio of the partition functions at two consecutive time steps using importance sampling

$$\frac{Z_{\theta,t}}{Z_{\theta,t+1}} = \mathbb{E}_{p_\theta(\mathbf{y}_{t+1})} \left[\exp(f_\theta(\mathbf{y}, t) - f_\theta(\mathbf{y}, t+1))\right] \tag{45}$$

$$\doteq \frac{1}{M} \sum_{i=1}^{M} \left[\exp(f_\theta(\mathbf{y}_{t+1,i}, t) - f_\theta(\mathbf{y}_{t+1,i}, t+1))\right], \tag{46}$$

where $\mathbf{y}_{t+1,i}$ are samples generated by progressive sampling. Starting from $t = T$, where $p_T(x)$ follows Gaussian distribution, we can compute $\log Z_{\theta,t}$ along the reverse path of the diffusion process, until we reach $t = 0$:

$$Z_{\theta,0} = Z_{\theta,T} \prod_{t=0}^{T-1} \frac{Z_{\theta,t}}{Z_{\theta,t+1}}. \tag{47}$$

In practice, since the ratio given by MCMC samples can vary across many orders of magnitude, it is more meaningful to estimate

$$\log Z_{\theta,0} = \log Z_{\theta,T} + \sum_{t=0}^{T-1} \log \frac{Z_{\theta,t}}{Z_{\theta,t+1}}. \tag{48}$$

Unfortunately, although equation 46 is an unbiased estimator of $Z_{\theta,t}/Z_{\theta,t+1}$, the logarithm of this estimator is generally a stochastic lower bound of $\log(Z_{\theta,t}/Z_{\theta,t+1})$ (Grosse et al., 2016). However, as we show below, this bound will gradually converge to an unbiased estimator of $\log(Z_{\theta,t}/Z_{\theta,t+1})$, as the number of samples becomes large. Specifically, let $A$ be the estimator in equation 46, $\mu$ be the true value of $Z_{\theta,t}/Z_{\theta,t+1}$. We have $\mathbb{E}[A] = \mu$, then by second order Taylor expansion,

$$\mathbb{E}[\log A] \doteq \mathbb{E}\left[\log \mu + \frac{1}{\mu}(A - \mu) - \frac{1}{2\mu^2}(A - \mu)^2\right] \tag{49}$$

$$= \log \mu - \frac{1}{2\mu^2}\mathrm{Var}(A). \tag{50}$$

By *law of large number*, $\mathrm{Var}(A) \to 0$ as $M \to \infty$, and thus $\mathbb{E}[\log A] \to \log \mu$. This is also consistent with the estimation curves in the right subfigure of Figure 7: since $\mathrm{Var}(A) \geq 0$, the estimation curve increases from below as the number of samples becomes larger. When the curve becomes stable, it indicates the convergence.

## B  EXPERIMENTAL DETAILS

**Model architecture.**  Our network structure is based on Wide ResNet (Zagoruyko & Komodakis, 2016). Table 5 lists the detailed network structures of various resolutions. The number of ResBlocks at every level $N$ is a hyperparameter that we sweep over. The values of $N$ for various datasets are listed in Table 6. Each ResBlock consists of two Conv2D layers. For the second Conv2D layer, we use zero initialization for the weights, and add a trainable channel-wise scaling parameter to the output. We remove the weight normalization, and use leaky ReLU (slope = 0.2) as the activation function in ResBlocks. Spectral normalization (Miyato et al., 2018) is used to regularize parameters in Conv2D layer, ResBlocks and Dense layer. For encoding time step $t$, we follow the scheme in (Ho et al., 2020). Specifically, the time step $t$ is first transformed into sinusoidal embedding, and then two Dense layers is added. The time embedding is added after the first Conv2D layer of each ResBlock.

**Training.**  We use Adam (Kingma & Ba, 2014) optimizer for all the experiments. We find that for high resolution images, using a smaller $\beta_1$ in Adam help stabilize training. We use learning rate 0.0001 for all the experiments. For the values of $\beta_1$, batch sizes and the number of training iterations for various datasets, see Table 6.

**Datasets.**  We use the following datasets in our experiments: CIFAR-10 (Krizhevsky et al., 2009), CelebA (Liu et al., 2018) and LSUN (Yu et al., 2015). CIFAR-10 is of resolution $32 \times 32$, and contains $50,000$ training images and $10,000$ test images. CelebA contains 202,599 face images, of which 162,770 are training images and 19,962 are test images. For processing, we first clip each image to $178 \times 178$ and then resize it to $64 \times 64$. For LSUN, we use church_outdoor and bedroom categories, which contains 126,227 and 3,033,042 training images respectively. Both categories contain 300 test images. For processing, we first crop each image to a square image of the smaller size among the height and weight, and then we resize it to $64 \times 64$ or $128 \times 128$. For resizing, we set antialias to True. We apply horizontal random flip as data augmentation for all datasets during training.

**Evaluation metrics.** We use FID and inception scores as quantitative evaluation metrics of sample quality. On all the datasets, we calculate FID and inception scores on 50,000 samples using the original code from Salimans et al. (2016) and Heusel et al. (2017).

Table 5: Model architectures of various solutions. $N$ is a hyperparameter that we sweep over.

(a) Resolution $32 \times 32$

| $3 \times 3$ Conv2D, 128 |
| --- |
| $N$ ResBlocks, 128
Downsample $2 \times 2$ |
| $N$ ResBlocks, 256
Downsample $2 \times 2$ |
| $N$ ResBlocks, 256
Downsample $2 \times 2$ |
| $N$ ResBlocks, 256 |
| ReLU, global sum
Dense 1 |

(b) Resolution $64 \times 64$

| $3 \times 3$ Conv2D, 128 |
| --- |
| $N$ ResBlocks, 128
Downsample $2 \times 2$ |
| $N$ ResBlocks, 256
Downsample $2 \times 2$ |
| $N$ ResBlocks, 256
Downsample $2 \times 2$ |
| $N$ ResBlocks, 256
Downsample $2 \times 2$ |
| $N$ ResBlocks, 512 |
| ReLU, global sum
Dense 1 |

(c) Resolution $128 \times 128$

| $3 \times 3$ Conv2D, 128 |
| --- |
| $N$ ResBlocks, 128
Downsample $2 \times 2$ |
| $N$ ResBlocks, 256
Downsample $2 \times 2$ |
| $N$ ResBlocks, 256
Downsample $2 \times 2$ |
| $N$ ResBlocks, 256
Downsample $2 \times 2$ |
| $N$ ResBlocks, 512
Downsample $2 \times 2$ |
| $N$ ResBlocks, 512 |
| ReLU, global sum
Dense 1 |

(d) Time embedding (temb)

| sinusoidal embedding |
| --- |
| Dense, leakyReLU |
| Dense |

(e) ResBlock

| leakyReLU, $3 \times 3$ Conv2D |
| --- |
| + Dense(leakyReLU(temb)) |
| leakyReLU, $3 \times 3$ Conv2D |
| + input |

Table 6: Hyperparameters of various datasets.

| Dataset | $N$ | $\beta_1$ in Adam | Batch size | Training iterations |
| --- | --- | --- | --- | --- |
| CIFAR-10 | 8 | 0.9 | 256 | 240k |
| CelebA | 6 | 0.5 | 128 | 880k |
| LSUN church_outdoor $64^2$ | 2 | 0.9 | 128 | 960k |
| LSUN bedroom $64^2$ | 2 | 0.9 | 128 | 760k |
| LSUN church_outdoor $128^2$ | 2 | 0.5 | 64 | 840k |
| LSUN bedroom $128^2$ | 5 | 0.5 | 64 | 580k |

## C  ADDITIONAL EXPERIMENTAL RESULTS

### C.1  FID SCORES OVER ITERATIONS

Figure 9 demonstrates FID scores computed on 2,500 samples every 15,000 iterations.

### C.2  LONG-RUN CHAIN SAMPLING WITH NUTS

As a further check, we use a No-U-Turn Sampler (Hoffman & Gelman, 2014) to perform the long-run chain sampling, with the same step size schedule obtained for HMC sampler. Figure 10 displays samples with different number of sampling steps. The samples remain realistic after $100k$ sampling steps in total and the FID score remains stable.

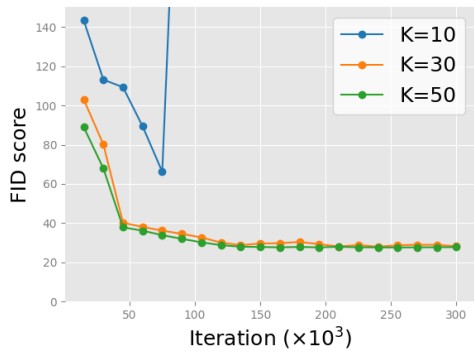

Figure 9: FIDs for different number of Langevin steps.

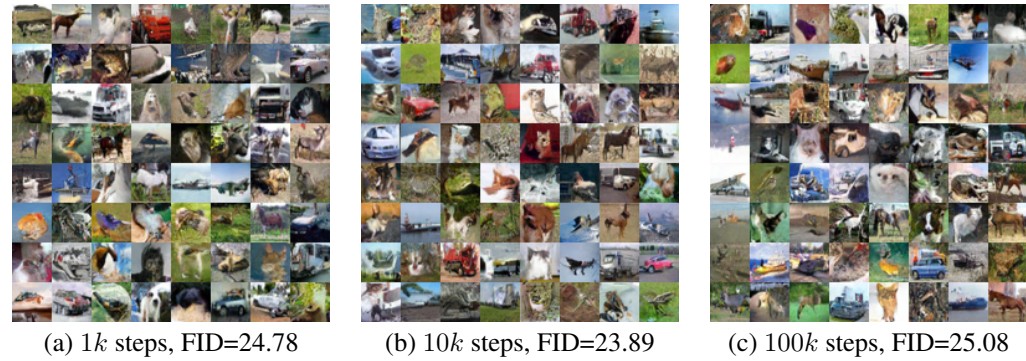

(a) $1k$ steps, FID=24.78     (b) $10k$ steps, FID=23.89     (c) $100k$ steps, FID=25.08

Figure 10: Long run chain samples with different total number of NUTS steps.

## C.3 ADDITIONAL INTERPOLATION RESULTS

Figures 11, 12 and 13 display more examples of interpolation between two generated samples on CelebA $64^2$, LSUN church_outdoor $128^2$ and LSUN bedroom $128^2$.

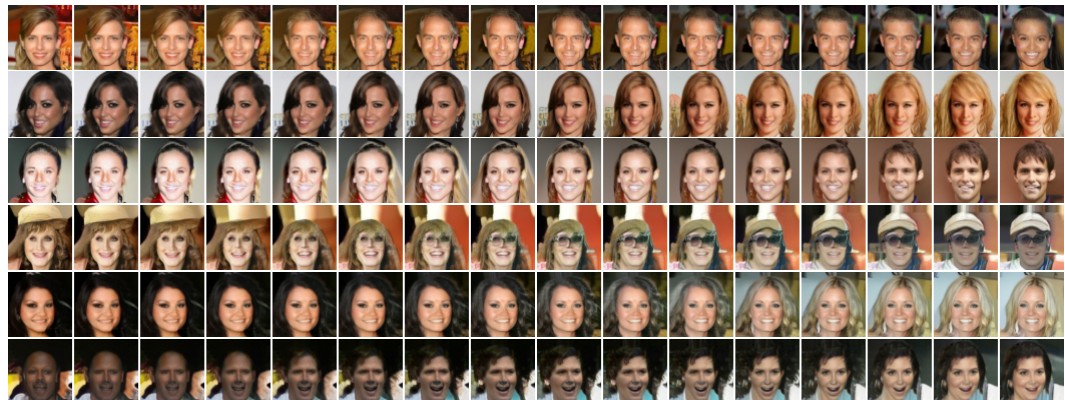

Figure 11: Interpolation results between the leftmost and rightmost generated samples on CelebA $64 \times 64$.

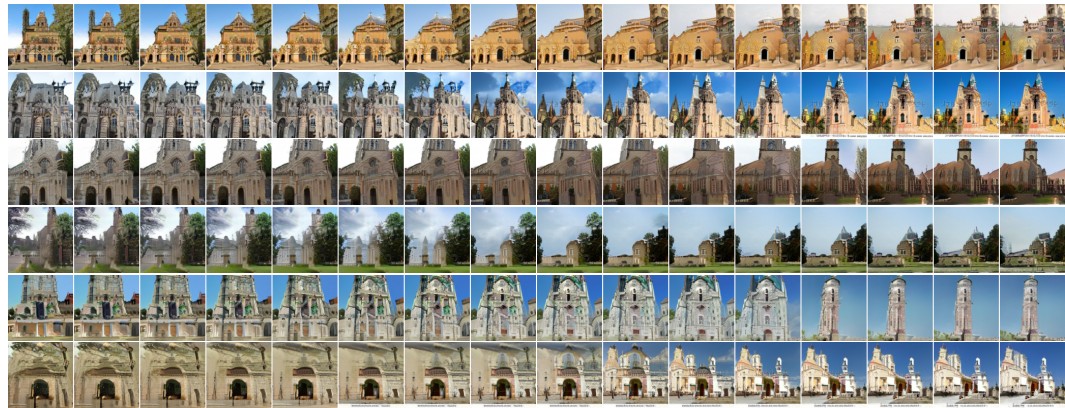

Figure 12: Interpolation results between the leftmost and rightmost generated samples on LSUN church_outdoor $128 \times 128$.

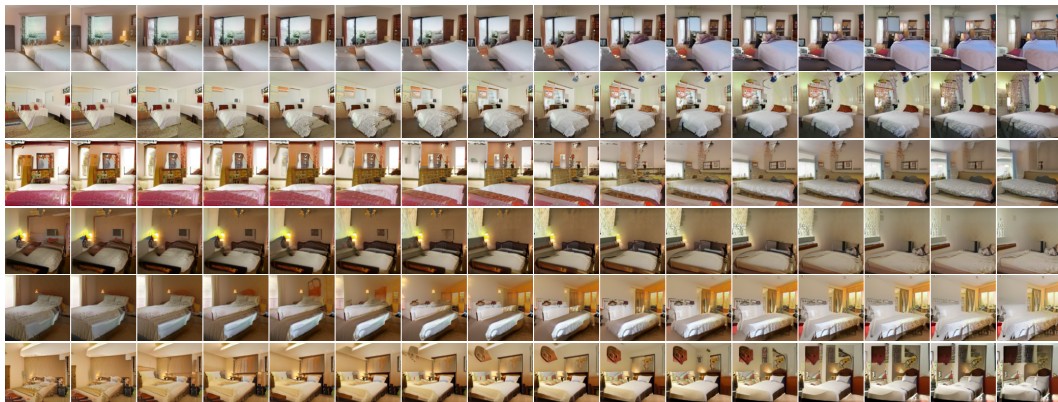

Figure 13: Interpolation results between the leftmost and rightmost generated samples on LSUN bedroom $128 \times 128$.

## C.4  ADDITIONAL IMAGE INPAINTING RESULTS

Figures 14 and 15 show additional examples of image inpainting on CelebA $64^2$ and LSUN church_outdoor $128^2$.

## C.5  ADDITIONAL UNCURATED SAMPLES

Figures 16, 17, 18, 19, 20 and 21 show uncurated samples from the learned models under *T6* setting on CIFAR-10, CelebA $64^2$, LSUN church_outdoor $128^2$, LSUN bedroom $128^2$, LSUN church_outdoor $64^2$ and LSUN bedroom $64^2$ datasets.

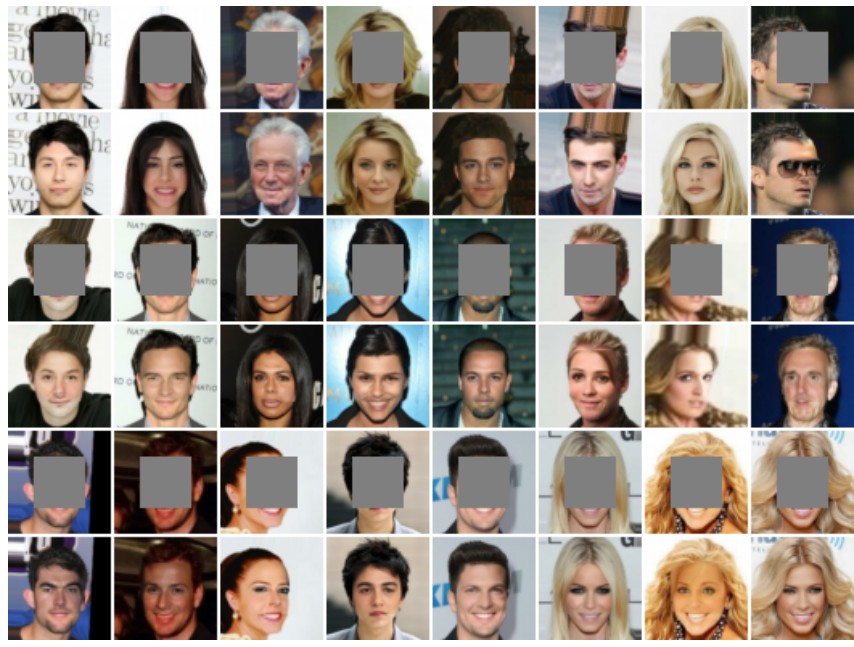

Figure 14: Image inpainting results on CelebA $64 \times 64$. *Top*: masked images, *bottom*: inpainted images.

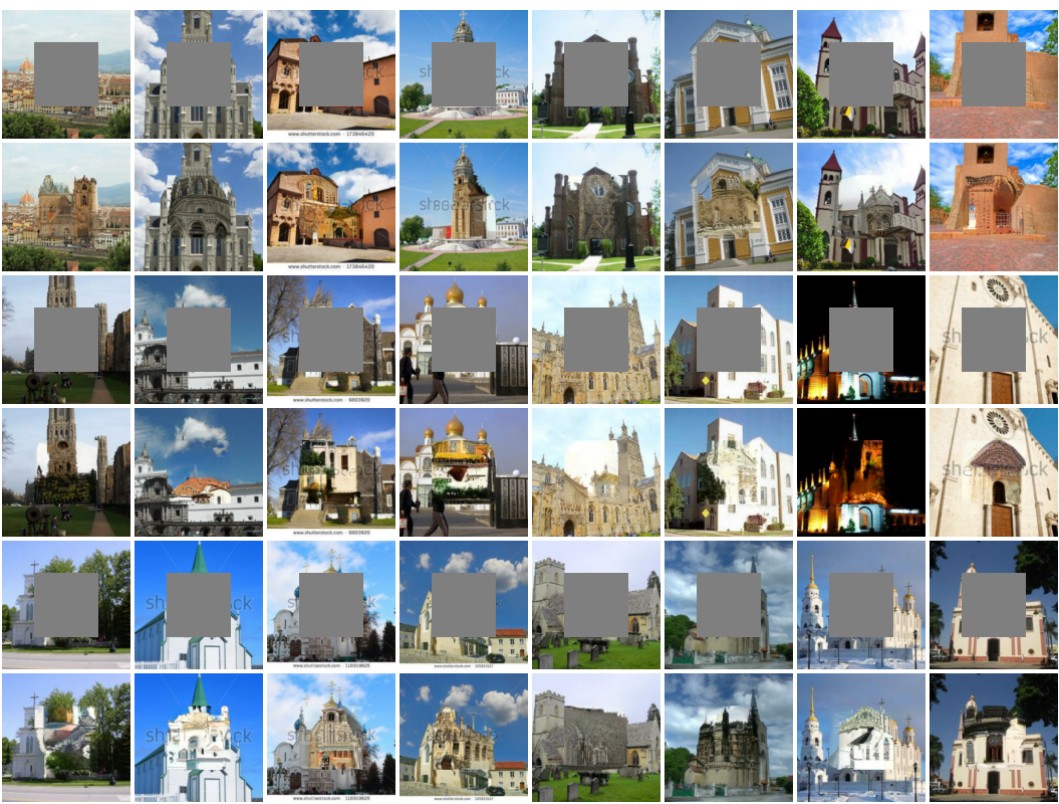

Figure 15: Image inpainting results on LSUN church_outdoor $128 \times 128$. *Top*: masked images, *bottom*: inpainted images.

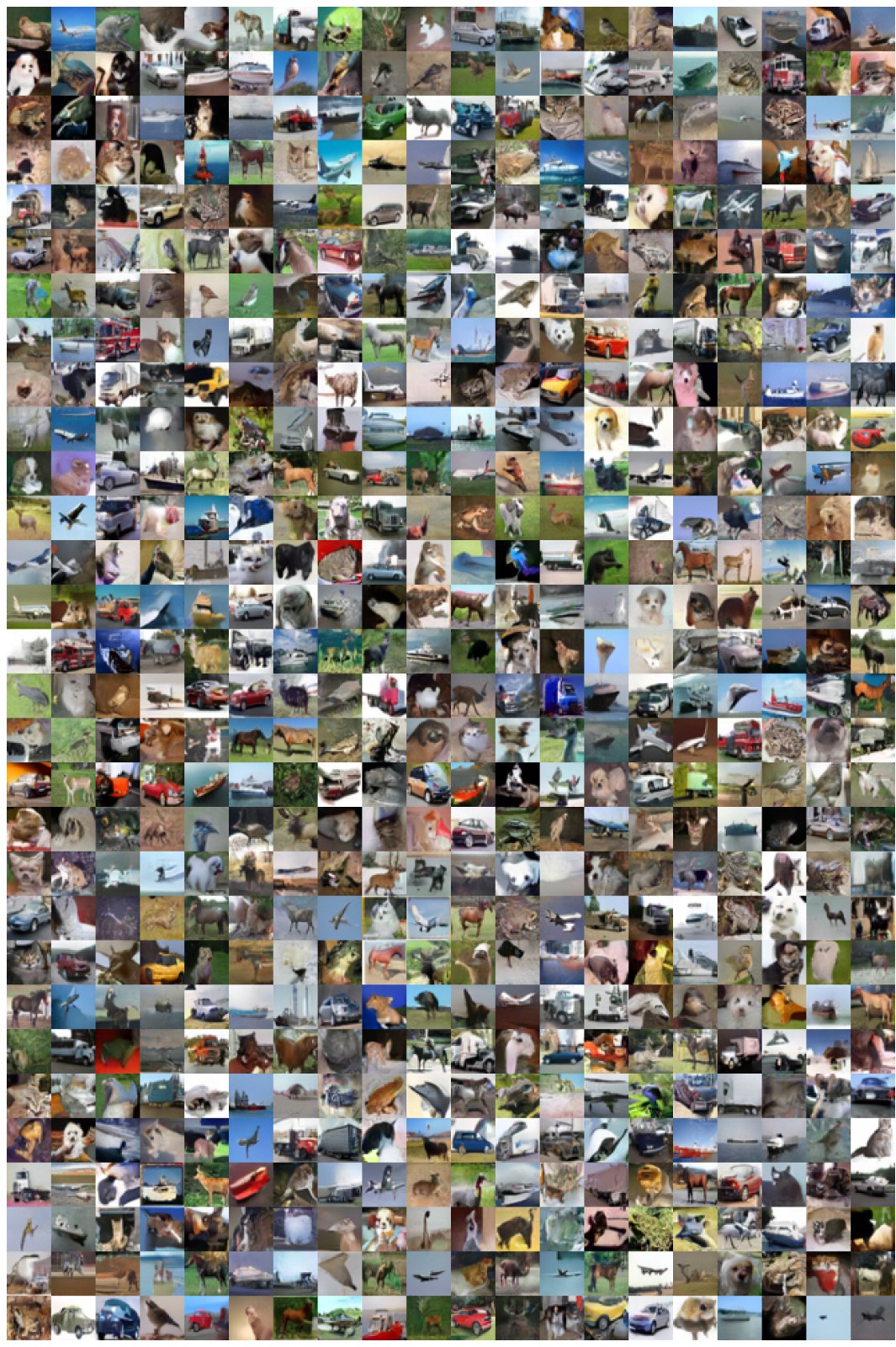

Figure 16: Generated samples on CIFAR-10.

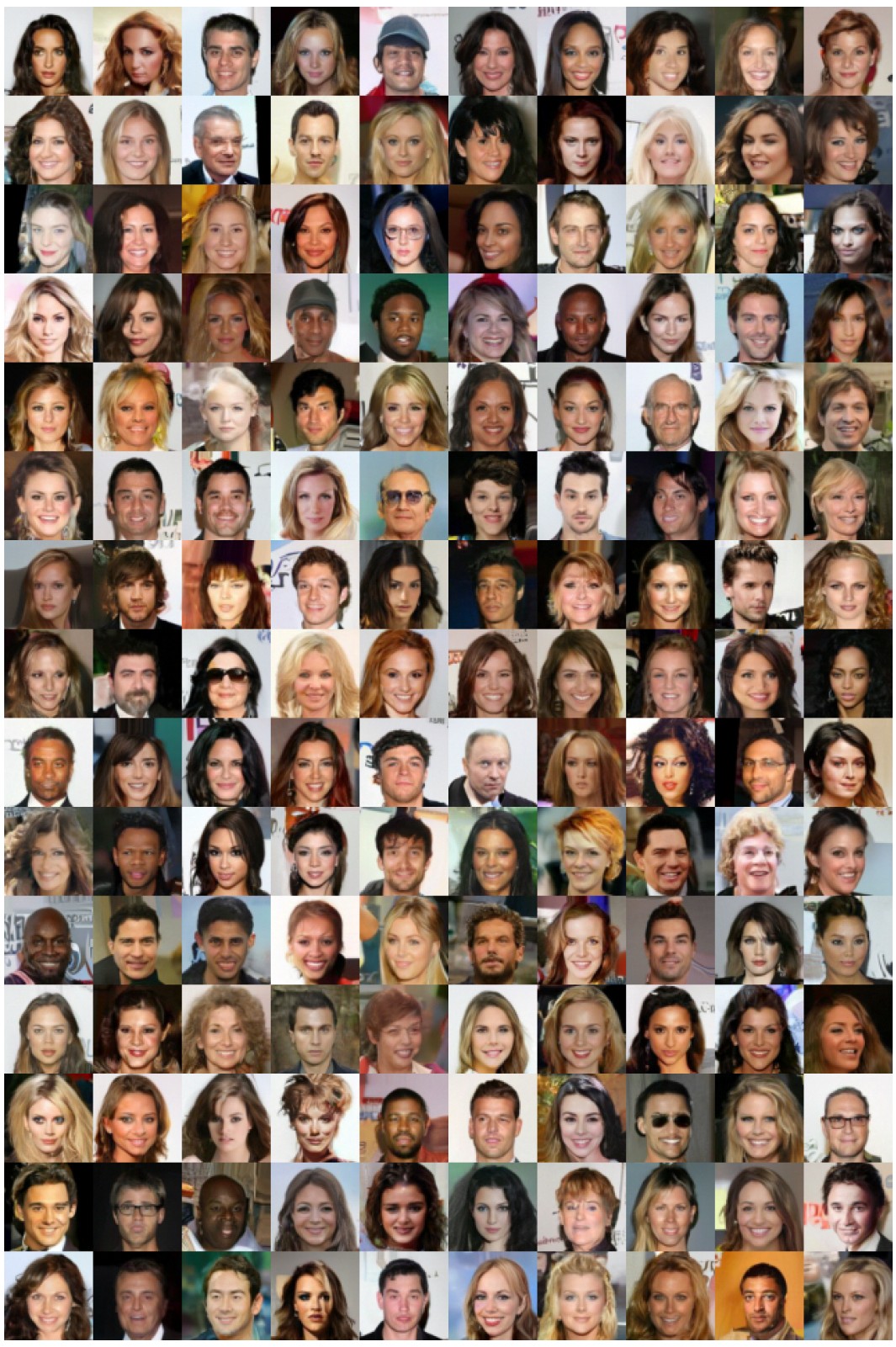

Figure 17: Generated samples on CelebA $64 \times 64$.

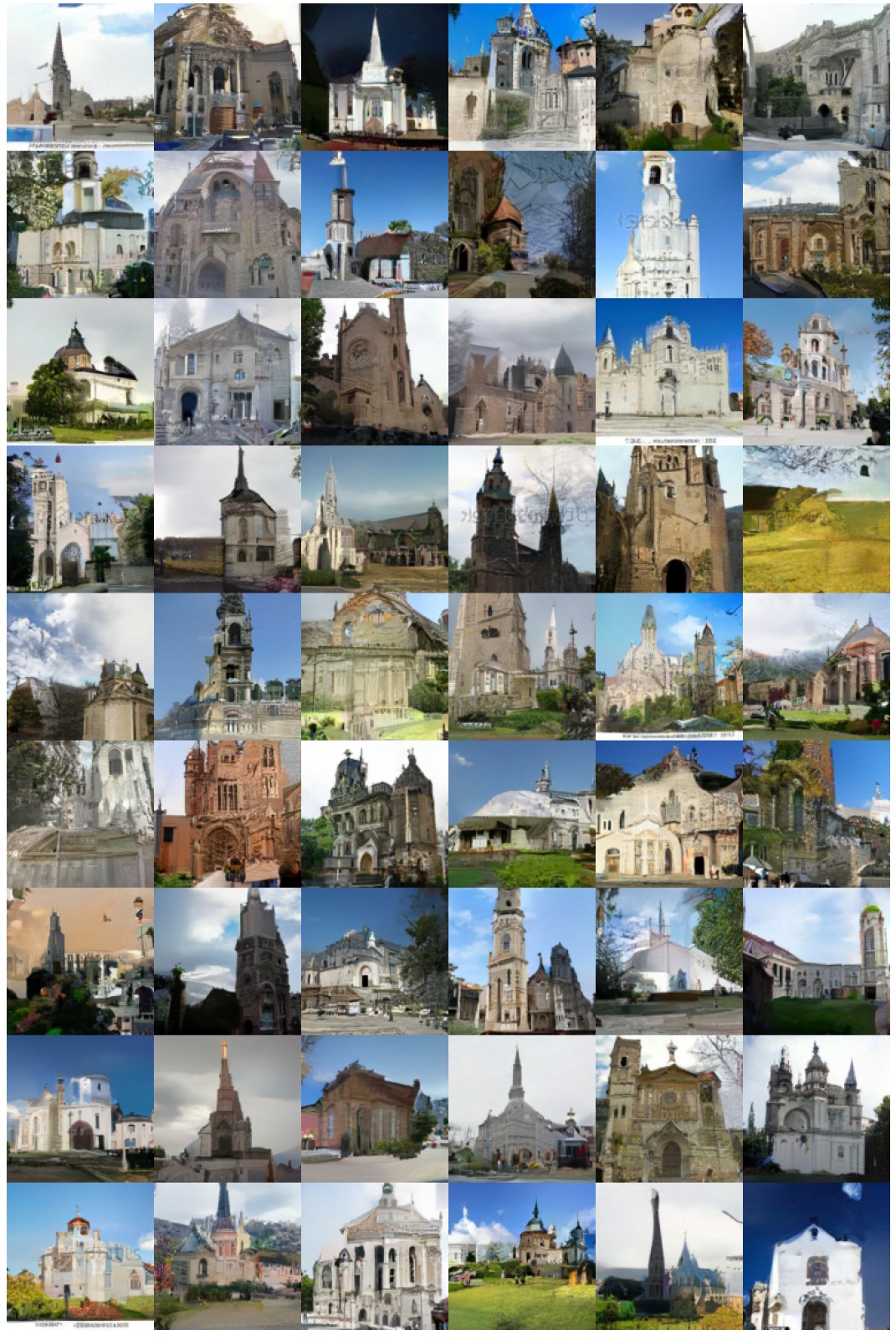

Figure 18: Generated samples on LSUN church_outdoor $128 \times 128$. FID=9.76

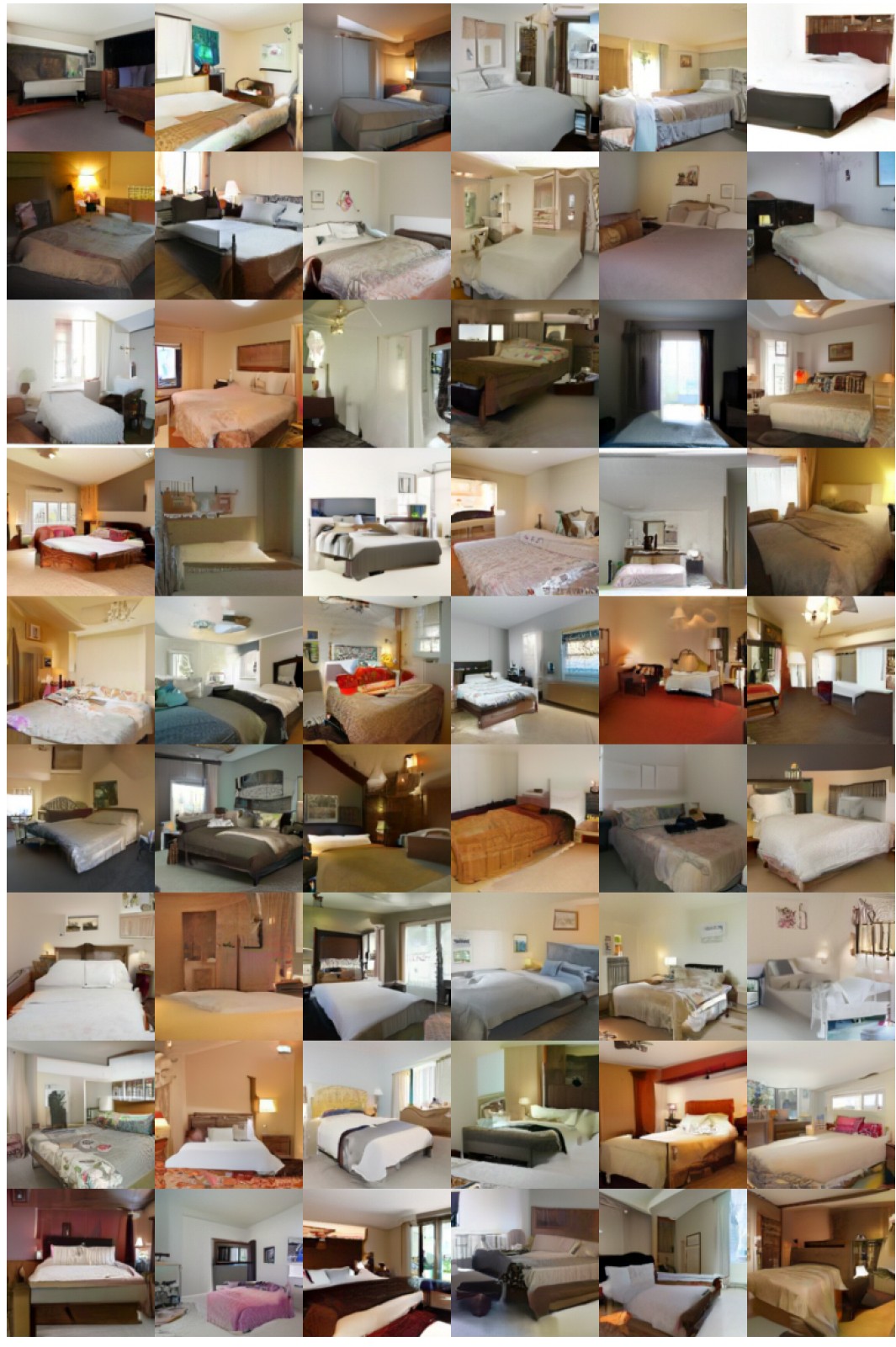

Figure 19: Generated samples on LSUN bedroom $128 \times 128$. FID=11.27

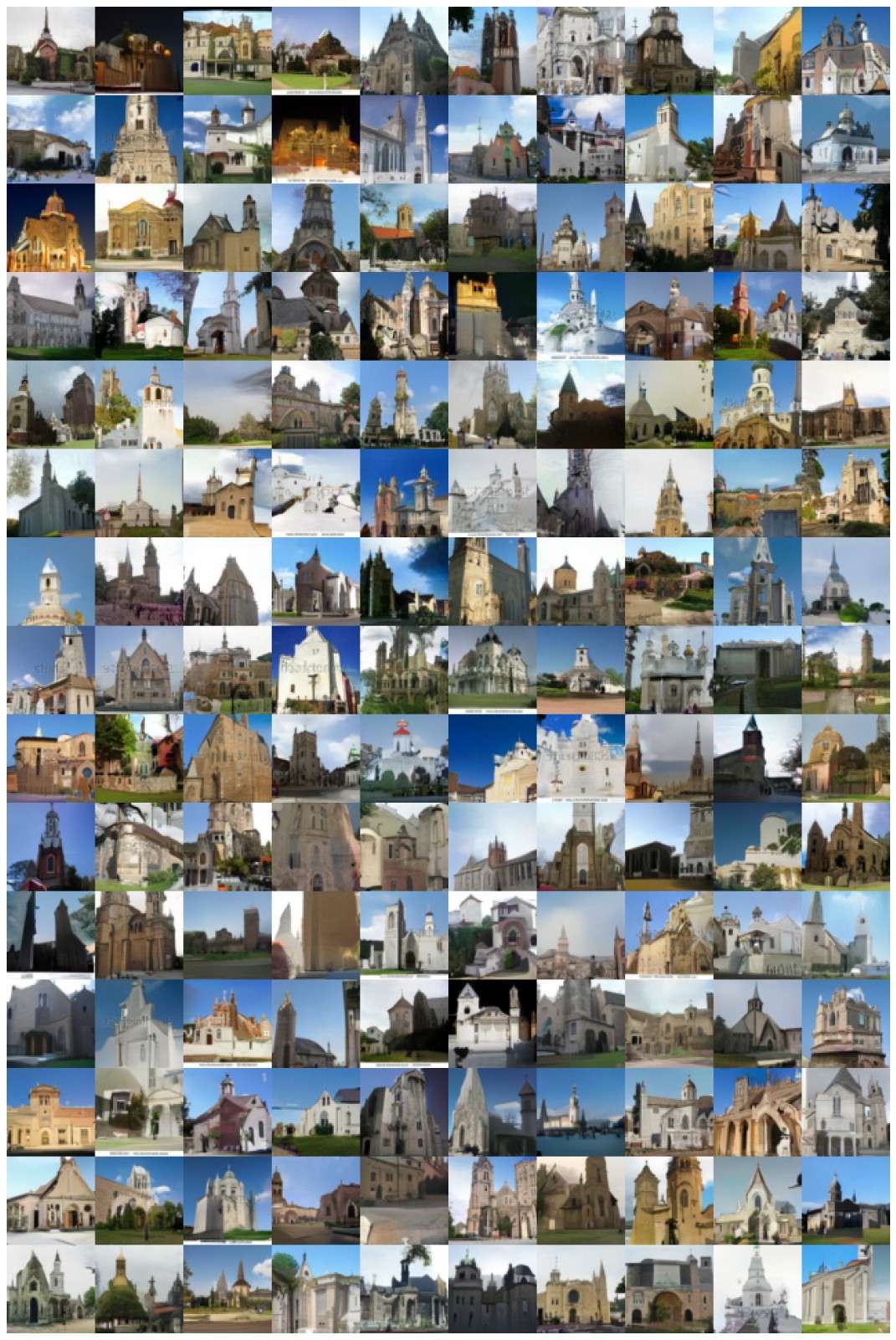

Figure 20: Generated samples on LSUN church_outdoor $64 \times 64$. FID=7.02

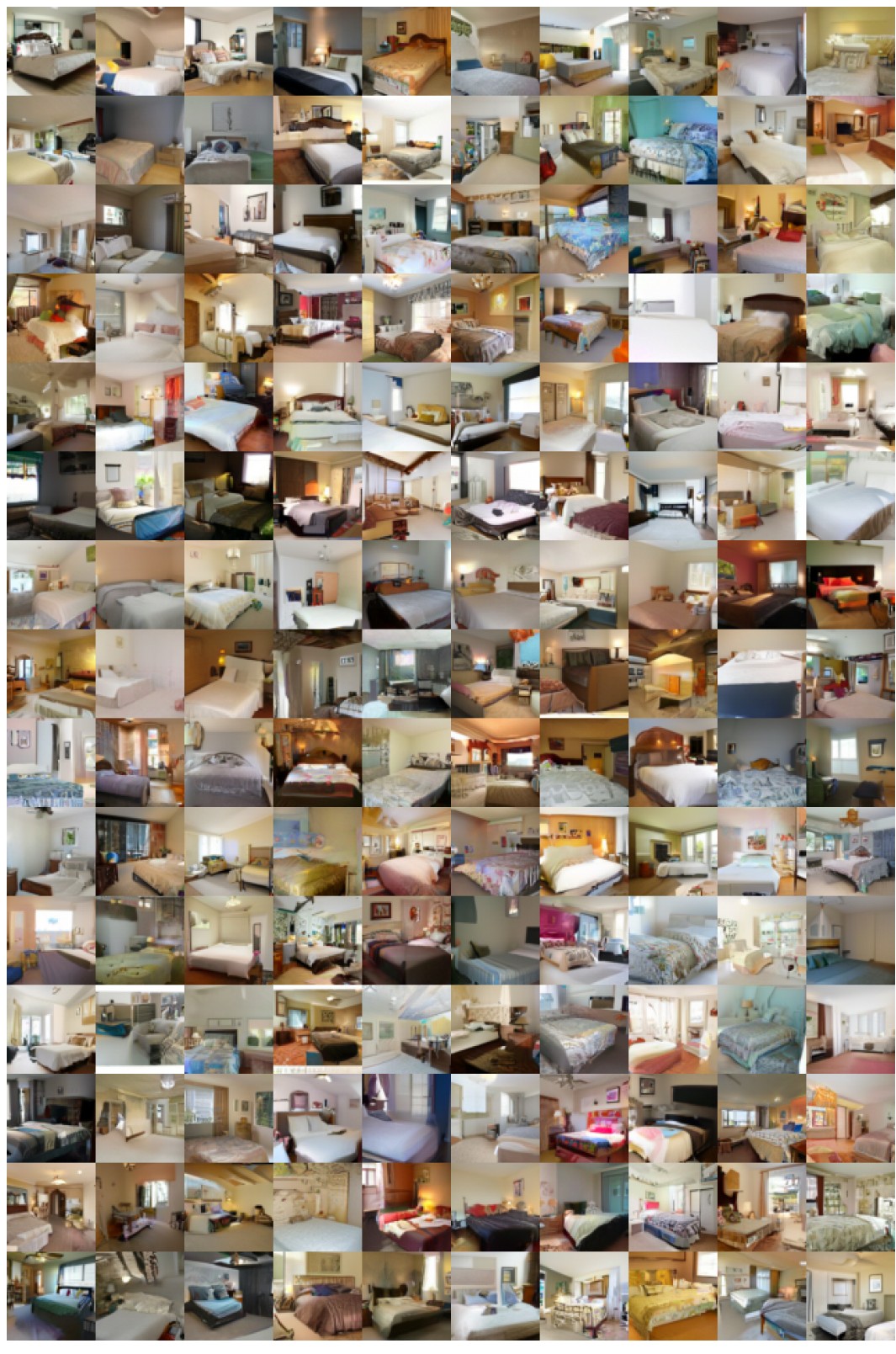

Figure 21: Generated samples on LSUN bedroom $64 \times 64$. FID=8.98

