# OpenReview forum: "Learning Energy-Based Models by Diffusion Recovery Likelihood"
_ICLR.cc/2021/Conference — ICLR 2021 Poster_

### Official Review · AnonReviewer3 · 2020-10-24
**Strong theory, experimental part is weaker**

**Rating:** 6
**Confidence:** 3

**Review:**

#### Summary of the paper :
The authors propose to learn the recovery likelihood of a sequence of Energy-Based Models (EBM) trained on increasingly noisy version of the datasets. The authors demonstrate that optimizing the recovery likelihood is more convenient than learning the actual marginal likelihood, as it leads to more stable MCMC sampling and provide high-fidelity samples. The authors show competitive generation performances on CIFAR-10, Celeb1 and LSUN datasets.

#### Pros :
* The problem of non-convergent MCMC sampling in EBM trained with marginal likelihood is well motivated and well illustrated.
* The authors proposition is well grounded in theory, and comes with rigorous and strong mathematical derivation.
* The paper is clear, well written, and well structured

#### Cons:
* The quantitative comparison (IS and FID) is performed only on the CIFAR-10 datasets (see detailed comments).
* The experimental evidences are not convincing.

#### Recommandation:
Given the strength of the theoretical derivation, I would tend to accept the paper (marginally above threshold). However, stronger experimental evidences / discussion concerning the comparison with [1] & [2] would be beneficial for a better recommandation.

#### Detailed comments:
* While the authors show qualitative sampling results on LSUN, CIFAR-10, and CelebA they are quantitatively comparing the proposed framework on CIFAR-10 only. Even if CIFAR is challenging for generative models, I think the quantitative comparison on other databases is missing. In particular the comparison with [1] and [2] is necessary as the main claim of the paper is to show improvement using recovery likelihood.

* The strength of the authors proposition is to provide a framework that could leverage less diffusion time steps than [2] while keeping good generation performance. Even if the authors clearly state such a theoretical advantage in paragraph 3.4, the experimental evidence is not convincing. Less diffusion time step (e.g. the case T=6, K=50) results in degraded FID and IS compared to [2]. In addition, the comparison between  T=1000, K=0 (DSM) versus [2] show a significant advantage in favor of [2] (they also use 1000 time step in [2]). What is the reason ? It suggests that optimizing the recovery likelihood is less efficient than optimizing the marginal one in diffusion problem...

* Comparable setup (i.e T=1000, K=0 (DSM) versus T=1000, K=0) lead also to degraded performance : Why score-matching approach leads to better performance than diffusion recovery ? The quality of the paper would be greatly improved with such a discussion.

#### Typos and suggestions to improve the paper
* The initials 'MLE' are not explicitly defined
* The Eq 3 should come with a demonstration (in Annex) or at least with a reference .
* Eq 9 should be an approximate sign as it is a Taylor expansion.
* The transition from Eq.9 to Eq.10 should be demonstrated (in Annex)
* Eq 13 : N —> \matcal{N}
* Algorithm 2, line 2: It seems that there is a typo : t <- T-1 to T? Is it : t <- T-1 to 0

[1] : Erik Nijkamp, Mitch Hill, Song-Chun Zhu, and Ying Nian Wu. On learning non-convergent short- run mcmc toward energy-based model. arXiv preprint arXiv:1904.09770, 2019b.

[2]: Jonathan Ho, Ajay Jain, and Pieter Abbeel. Denoising diffusion probabilistic models. arXiv preprint arXiv:2006.11239, 2020.

---

> ### Author Response · Authors · 2020-11-25
> **Thank you for your insightful comments**
>
> Thank you for your time and constructive comments. We will reply to your points in order.
>
> - Quantitative comparison.
>
> We have added a new Table 3 to include quantitative comparisons on the CelebA 64x64 dataset. Our method achieves FID of 5.98, which is competitive to the state-of-the-art GAN-based method. We also report FID scores on LSUN datasets in Appendix C.4 (in the caption of generated samples). According to Tables 1 and 3, our method outperforms [1] by a large margin on both CIFAR-10 and CelebA datasets. As for comparison to [2], we would like to emphasize that our goal is to learn explicit energy-based models, which have a number of promising applications and desirable properties as shown by [3, 4], whereas [2] parametrizes and learns implicit score models that may not correspond to the gradient of an energy. In the ablation study, we include a result of (T=1000, K=0 (DSM)), where we use exactly the same implementation as in [2], and replace the implicit score model by the gradient of the explicit energy-based model. For this explicit energy-based model we consider, learning by maximizing recovery likelihood (T=6, K=50) achieves better performance than learning by [2] (T=1000, K=0 (DSM)). We have added more explanation to the ablation study (section 4.2).
>
> - About experimental evidence.
>
> We would like to clarify that the difference between [2] and (T=1000, K=0 (DSM)) is only about the parametrization of the score function. [2] directly parametrizes the score function with a U-net  network without explicitly parameterizing the energy function, whereas (T=1000, K=0 (DSM)) parametrizes the score function with the gradient of an explicit energy function. (T=1000, K=0 (DSM)) uses the same training objective and procedure as [2]. The gap between [2] and (T=1000, K=0 (DSM)) indicates that [2] performs worse when it is applied to learning an explicit energy function instead of a score function.  For the explicit energy-based model, learning by maximizing recovery likelihood (T = 6, K =50) performs better than learning by [2] or denoising score matching. This result also motivates a future work about searching for an optimal explicit energy function that is suitable for the objective of [2].
>
> - About comparison of (T=1000, K=0 (DSM)) and (T=1000, K=0).
>
> In section 3.4 we have shown that the training objectives of the two settings agree with each other. In the revision we add a new section Appendix A.4 to further show that the learning gradient of maximizing normal approximation (T=1000, K=0 (DSM)) is roughly the same as the one of maximizing recovery likelihood (T=1000, K=0). Table 2 shows that (T=1000, K=0 (DSM)) and (T=1000, K=0) achieve similar sample quality in terms of quantitative metrics, where (T=1000, K=0 (DSM)) gives a slightly better FID score while (T=1000, K=0) achieves a slightly better inception score. This result further verifies that the training procedures of (T=1000, K=0 (DSM)) and (T=1000, K=0) are consistent with each other.
>
> - About typos and suggestions.
>
> We have corrected typos and followed your suggestions to revise the paper. Thank you.
>
>
> [1] Erik Nijkamp, Mitch Hill, Song-Chun Zhu, and Ying Nian Wu. On learning non-convergent short- run mcmc toward energy-based model. arXiv preprint arXiv:1904.09770, 2019.
>
> [2] Jonathan Ho, Ajay Jain, and Pieter Abbeel. Denoising diffusion probabilistic models. arXiv preprint arXiv:2006.11239, 2020.
>
> [3] Will Grathwohl, Kuan-Chieh Wang, Jo ̈rn-Henrik Jacobsen, David Duvenaud, Mohammad Norouzi, and Kevin Swersky. Your classifier is secretly an energy-based model and you should treat it like one. arXiv preprint arXiv:1912.03263, 2019.
>
> [4] Yilun Du and Igor Mordatch. Implicit generation and generalization in energy-based models. arXiv preprint arXiv:1903.08689, 2019.

---

### Official Review · AnonReviewer2 · 2020-10-27
**Very nice method!**

**Rating:** 7
**Confidence:** 3

**Review:**

The paper proposed a novel method to train EBMs based on diffusion recovery likelihood. It constructs a sequence of noisy version of data and learn a conditional between consecutive noisy pairs. Compare to working with the likelihood directly, doing so makes the training much easier. Besides, even using a potentially non-convergent MCMC for gradient estimation, it still leads to a well-behaved energy potential, unlike EBMs trained via maximising the likelihood directly.

## Pros

1. The paper is well-written and easy to understand.
2. The method is novel and solves the training difficulty issue for EBMs neatly.
3. The analysis of the normal approximation and how it leads to choose step sizes in the Langevin dynamics is neat.
4. The paper discusses related work well while motivating the method and also makes interesting connections to VI and score matching in Section 3.4.

## Cons

1. Only applications on image generations are conducted.

## Questions

1. Does the method work only with the specific architecture here, especially the Transformer sinusoidal position embedding?
2. It says T = 6 and K = 10 leads to unstable training in Section 4.1. Does the optimization diverge or something? It would be good to see some trace plots for the training procedure.
3. Why do you use the specific HMC setup for Section 4.2? Specifically, using a leapfrog step of 2 can lead to issues in which the MCMC only explore local modes. I would like to see how a long-run NUTS works here. A well-behaved potential should work in both settings.

## Discussions

1. I'm quite glad to see Section 4.2 that shows the long-run MCMC works on the trained EBMs, even with (potentially) biased gradients for training. Is it because we only need relative short-run MCMC for convergence when using recovery likelihood so that the bias is actually small?
2. How does the method relate to amortized MCMC?

## Related work

There are a few more related works that the authors may consider including.

[1] proposed a similar idea of making the modelling task easier using a sequence of targets (for EBMs), but achieved by density ratio estimation.
[2] uses coupled MCMC to resolve the biased gradient issue when using CD.

[1] Rhodes, Benjamin, Kai Xu, and Michael U. Gutmann. "Telescoping Density-Ratio Estimation." Advances in Neural Information Processing Systems. 2020.
[2] Qiu, Yixuan, Lingsong Zhang, and Xiao Wang. "Unbiased Contrastive Divergence Algorithm for Training Energy-Based Latent Variable Models." International Conference on Learning Representations. 2019.

---

> ### Author Response · Authors · 2020-11-25
> **Thank you for your insightful comments**
>
> Thank you for your kind words and thoughtful comments. We will reply to your points in order.
>
> - About applications other than image generations.
>
> We have added a new experiment of image inpainting in section 4.1 and Appendix C.3. Specifically, the inpainting is achieved by running Langevin dynamics progressively on the masked pixels while keeping the observed pixels fixed at decreasingly lower noise levels. We demonstrate that the learned model is capable of realistic and semantically meaningful image inpainting on CelebA and LSUN datasets. We will further explore other applications of energy-based models using our method in the future work.
>
> - About the model architecture
>
> In the early experiments, we also tried to use some other architectures. For example, for the energy function, we once used a 5-layer convolutional neural network instead of the residual network, which works reasonably well on simple datasets such as SVHN and MNIST. For the embedding of the timestep, we have tried to learn a separate top fully connected layer for each time step instead of using the Transformer sinusoidal position embedding. As [1] pointed out, such parametrization is optimal if the network is powerful enough. In practice, it results in worse but still reasonable sample quality (on CIFAR-10, FID 18.34, inception 8.10).
>
> - About T = 6 and K = 10. Does the optimization diverge or something?
>
> Yes, in this case the optimization diverges. The loss explodes to a very large negative value and samples are not realistic any more. This is because with limited sampling steps, the sampling chains fail to reach the local modes of the current model, and therefore for the learning gradient (equation 3),   $E_{\mathbf{x} \sim p_\theta}[\frac{\partial }{\partial \theta} f_\theta(\mathbf{x})]$ cannot keep pace with $E_{\mathbf{x} \sim p_\text{data}}[\frac{\partial }{\partial \theta} f_\theta(\mathbf{x})]$. Following your suggestion, we have included a plot of FID scores over iterations in Figure 7, where we compare the FID scores of models trained with different sampling steps.
>
> - About the sampler.
>
> We apply HMC sampler for long-run chain analysis as it can generally explore better than Langevin dynamics. Following your suggestion, we have performed long-run chain sampling using No-U-Turn sampler (NUTS) and included the results in section 4.3 and Appendix C.1. We apply the same step size schedule obtained for HMC to NUTS. The long-run chain samples are still realistic after 100k steps in total and the FID score remains stable, indicating the validity of the learned potential.
>
> - The reason why long-run MCMC works. Is it because we only need relative short-run MCMC for convergence when using recovery likelihood so that the bias is actually small?
>
> Yes, you are right. The reason is that the conditional distribution defined by recovery likelihood has a much simpler energy landscape, where the short-run MCMC has higher chance to converge, so that the model could be less biased.
>
> - Relation to amortized MCMC.
>
> Compared to short-run MCMC [2], our method can be considered amortizing the MCMC sampling by a sequence of models at different time steps. Specifically, during testing, the MCMC sampling at time step t is initialized using samples obtained from t+1, and therefore models from t+1 to T can be considered an approximation network for the samples at t. Such amortization allows us to use less sampling steps than [2] during training.
>
> - About references.
>
> We have included and discussed the two references that you mentioned in sections 1 and 2. Thanks for pointing them out.
>
> [1] ICE-BeeM: Identifiable conditional energy-based deep models
>
> [2] Erik Nijkamp, Mitch Hill, Song-Chun Zhu, and Ying Nian Wu. On learning non-convergent short- run mcmc toward energy-based model. arXiv preprint arXiv:1904.09770, 2019.

---

### Official Review · AnonReviewer1 · 2020-11-02
**Interesting approach for training EBMs as a sequence of conditional EBMs**

**Rating:** 7
**Confidence:** 4

**Review:**

This paper describes training a sequence of conditional EBMs (inspired by  Ho et al. (2020)) instead of training unconditional EBMs.  Each conditional energy describes the probability of recovering x, given its noisy version \hat{x}. The noisy version of x can be described as a normal distribution centered at x, so the condition EBM has an additional term ||x - \hat{x}||^2, which constrains the Langevin dynamics to remain in the vicinity of \hat{x}, so it converges faster!
The inference is that starting at white noise x_0, it defines a conditional EBMs given x_0, runs Langaving dynamics to sample from P(x|x_0) defined by the conditional EBMs, and then use the sample as the evidence of another conditional EBMs.

I assume the main advantage of this training over Ho et al. (2020) is having an energy-based model that can be used in other applications, so that would be nice to see the performance of trained EBMs on some applications other than image generation such as image inpainting and robust classification as discussed in Du and Mordatch (2019).

In general, I believe that this is an exciting paper and an important step towards training better EBMs.

In connection to score matching, Saremi et al. (2018) and Saremi and Hyvarinen (2019) should be cited as well.

Saremi et al. (2018), Deep Energy Estimator Networks.
Saremi and Hyvarinen (2019), Neural Empirical Bayes.

typo: "score-based based methods" in Section 4.1

Alg2: for t \gets T - 1 to 0 do

---

> ### Author Response · Authors · 2020-11-25
> **Thank you for your insightful comments**
>
> Thank you for your time and your kind words about our work. We will reply to your points in order.
>
> - About applications of energy-based models
>
> As you suggested, we have added a new experiment on image inpainting in section 4.1 and Appendix C.3. Specifically, given a masked image and the corresponding mask, we first obtain a sequence of perturbed masked images at different noise levels. The inpainting is achieved by running Langevin dynamics progressively on the masked pixels while keeping the observed pixels fixed at decreasingly lower noise levels. We demonstrate that the learned model is capable of realistic and semantically meaningful image inpainting on CelebA and LSUN datasets. As for the robust classification experiment in [1], it has been used for evaluating conditional EBMs given labels, whereas we mainly focus on learning unconditional EBMs in this work. However, we totally agree with you that it would be an interesting future direction to learn conditional EBMs by our method and apply the learned models to robust classification.
>
> - About references
>
> Thanks for the two references, which are indeed very relevant. We have cited and discussed them in section 1 and section 3.4.
>
> - About typos
>
> We have fixed them in the revision. Thanks for pointing them out.
>
> [1] Yilun Du and Igor Mordatch. Implicit generation and generalization in energy-based models. arXiv preprint arXiv:1903.08689, 2019.

---

### Author Response · Authors · 2020-11-25
**Revised version of the submission**

We thank the reviewers for their insightful comments and careful reviews of our paper. We have uploaded a revised version of the paper which includes suggestions and addresses concerns of the reviewers. We summarize the revision as follows:

1. Applications of energy-based models
- A new image inpainting experiment in section 4.1 and Appendix C.3.
2. Evaluation metrics
- A new Table 3 including FID scores on CelebA; Report of FID scores on LSUN dataset in Appendix C.4.
- A new Figure 7 reporting FID scores over iterations.
3. Ablation study and comparisons
- Revised section 4.2 that clarifies various settings of ablation study
 - A new proof for the consistency of the learning gradients of normal approximation and recovery likelihood (Appendix A.5)
4. Long-run sampling
- A new long-run chain sampler in section 4.3 and Appendix C.1.
5. Include references pointed out by reviewers.
6. Fix typos and add derivations as suggested by reviewers.

---

### Decision · Program_Chairs · 2021-01-07
**Final Decision**

**Decision:**

Accept (Poster)

**Comment:**

The paper presents an interesting model to sample from the Gibbs distribution using diffusion based method. The theory is interesting and it is related well to the current research in the field. All reviewers agree that this is a noteworthy contribution to ICLR.